# Recognition of anion-water clusters by peptide-based supramolecular capsules

Victoria López-Corbalán [1], Alberto Fuertes [1], Antonio L. Llamas-Saiz [2], Manuel Amorín [1] & Juan R. Granja [1] ✉

The biological and technological importance of anion-mediated processes has made the development of improved methods for the selective recognition of anions one of the most relevant research topics today. The hydration sphere of anions plays an important role in the functions performed by anions by forming a variety of cluster complexes. Here we describe a supramolecular capsule that recognizes hydrated anion clusters. These clusters are most likely composed of three ions that form hydrated C3 symmetry complexes that are entrapped within the supramolecular capsule of the same symmetry. The capsule is made of self-assembled α,γ-cyclic peptide containing amino acid with by five-membered rings and equipped with a tris(triazolylethyl)amine cap. To recognise the hydrated anion clusters, the hexapeptide capsule must disassemble to entrap them between its two subunits.

Anion recognition is a biologically and technologically relevant process in which ad hoc designed receptors are able to bind to a specific ion triggering a specialised function[1,2]. In the last years, a large number of new synthetic receptors have been developed for this purpose[3,4]. The design principles used in this development are generally dominated by directional non-covalent interactions, such as halogen, chalcogen or hydrogen bonds, among others, rather than by purely electrostatic ones, as they generally provide better selectivity[5,6]. In this regard, nature is able to differentiate biologically relevant ions of similar properties, such as fluoride or chloride, to carry out selective cross-membrane transport efficiently[7,8]. The strong hydration of anions is one of the main obstacles to their recognition due to the additional energetic penalty paid for their desolvation during host–guest binding. In some cases, the ion selectivity in aqueous media does not depend as much on the relative size or charge of the ions, but on the size of their corresponding hydrates[9,10]. In fact, some authors have proposed that the selectivity in fluoride transport by FLuC protein (Fluoride-specific ion channel) could be explained in a similar way to the $K^+/Na^+$ selectivity of sodium ion channels[11]. Additionally, the greater solubility of negatively charged proteins may be related to stronger hydration free energies for negatively charged groups on protein surfaces than their cationic counterparts[12]. Therefore, water-anion clusters represent an important chemical complex that is not yet fully understood despite their biological relevance. Several experimental and theoretical studies have been carried out to unmask the behaviour of hydrated halide clusters or other ions[13–16]. Most of these clusters are formed by a single ion surrounded by a variable number of water molecules, although polyanionic clusters have also been reported[17]. The recognition of two or more anions is critical to many biological processes, such as chloride transport[18]. One important approach to studying this type of cluster is by analysing those generated in confined spaces[19,20]. Thus, several water clusters have been characterized by studying those formed in small protein pockets or other nanostructured materials. In this work we describe a class of hydrated polyanionic clusters, most likely three, embedded between two cyclic peptide subunits of a supramolecular dimeric capsule.

Supramolecular capsules are dynamic molecular assemblies composed of two or more self-complementary units held together by non-covalent interactions that, upon assembling, form an empty closed system whose internal cavities are useful tools in host–guest chemistry[21,22]. They can be considered to create a type of unconventional molecular environment in which trapped molecule(s) can exist. A wide variety of supramolecular capsules have been prepared

[1]Centro Singular de Investigación en Química Biolóxica e Materiais Moleculares (CIQUS) and Organic Chemistry Department, Universidade de Santiago de Compostela, Santiago de Compostela, Spain. [2]Unidad de Rayos X; Área de infraestructuras de Investigación, RIAIDT Edificio CACTUS, Universidade de Santiago de Compostela, 15782 Santiago de Compostela, Spain. ✉e-mail: juanr.granja@usc.es

with complementary or opposite functions, ranging from large to small, from charged to neutral or from open to closed systems. They exhibit unique properties, such as notable spectrochemical, electrochemical or magnetic effects. Therefore, they are useful tools for the molecular recognition of a wide variety of anionic, cationic or neutral guests (including gases) and are also able to catalyse a number of reactions leading to unusual products in a remarkably efficient way[23–26]. Recently, we have shown that flat-shaped cyclic peptides[27,28] equipped with a molecular cap (porphyrin moiety) provide a hydrogen-bonded dimeric receptor that was able to entrap long linear 4,4′-bipyridyl derivatives (Fig. 1)[29]. Here, we have created a smaller variant topped with a tris-triazolylamine component that is able to recognize water-anion clusters. To do this, the amide protons involved in the hydrogen bonds that supramolecularly bind both receptor subunits participate in the recognition of the hydrated anion. These clusters sandwiched between the two cyclic peptide components resemble a molecular burger, in which the anionic complexes are intercalated, like meat patty and cheese, between the two subunits that play the role of a molecular bun. The isobutyl pendants of the Leu side chain are cross interdigitated providing a hydrophobic supramolecular complex that can facilitate ion transport across lipid membranes.

## Results and discussion

Recently, we have prepared the cyclic hexapeptide bearing three propargyl chains attached to the γ-residues (**CP1**) that are pointing opposite to the dimerization plane (**D1**)[30–32]. These groups not only prevent stacking of cyclic peptides across that face, thereby restricting the assembling process to discrete dimers rather than nanotubes, but can also act as valuable reactive elements for the incorporation of other functional moieties at the entrance of the dimer (Fig. 1). For example, Sonogashira cross-coupling provided derivatives bearing ortho- or meta-oriented pyridine moieties[31]. The resulting peptides were able not only to encapsulate Xe atoms in the tubular structure of stacked dimers but also to efficiently transport ion pairs in model membranes[30]. Additionally, the acetylene moiety was used in a copper-catalysed alkyne-azide cycloaddition reaction to incorporate a variety of oligoethylene glycol pendants to generate membrane spanners[32]. Here, we describe the preparation of self-assembled cavitands (Fig. 2) covered by a tris-triazolyl motif attached to a central core that not only plays the role of a molecular cap but also acts as a binding site for the recognition of small anions[33–38].

Taking into account the aforementioned objectives, the cyclic peptide **CP1** was synthesized using the protocol described previously (Fig. 2)[30]. For the cap incorporation, **CP1** was subjected to a triple click

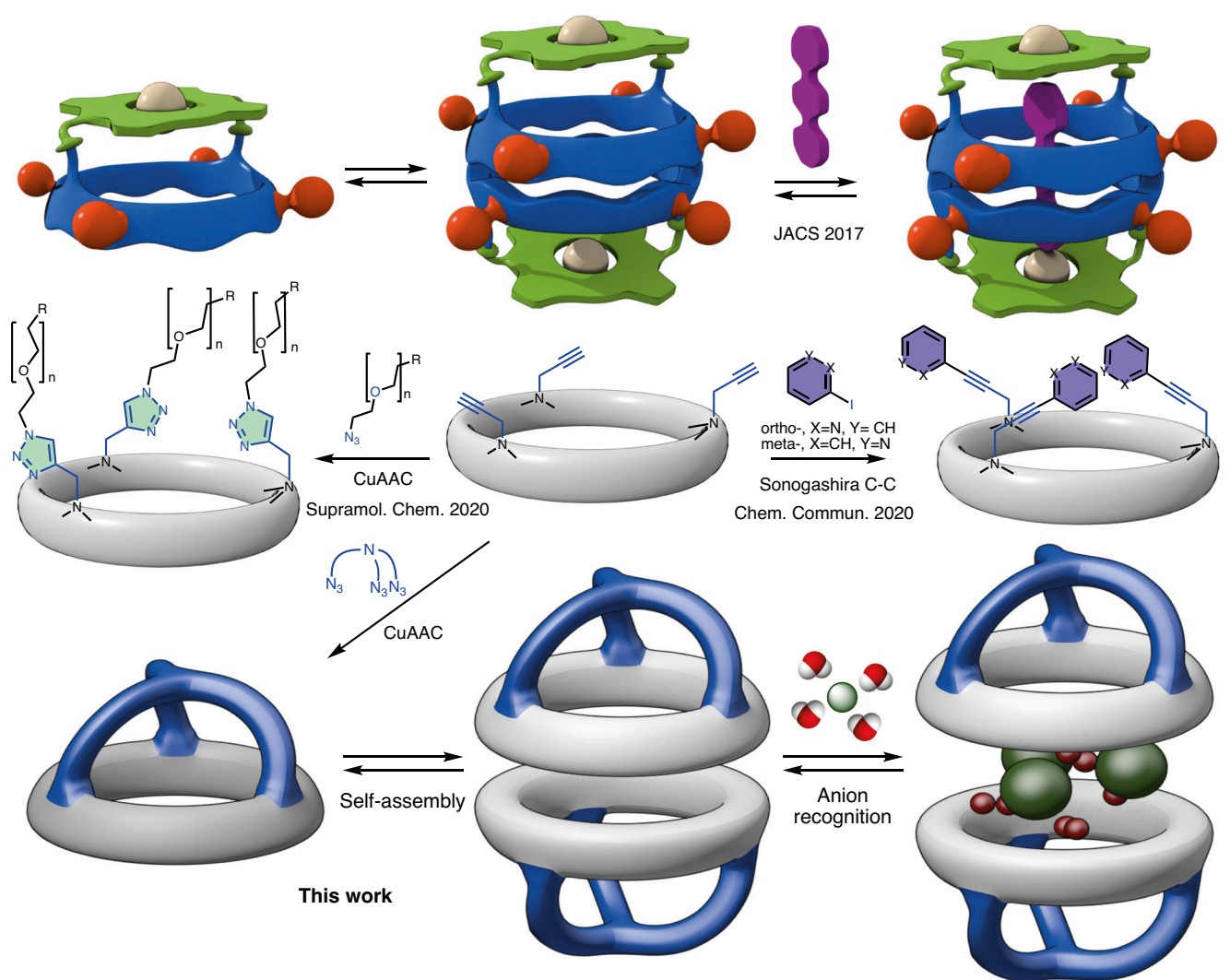

**Fig. 1 | Previously reported and proposed model of supramolecular containers based on α,γ-cyclic peptides.** Top: cyclic octapeptide (blue) topped with a porphyrin moiety (green) used in the recognition of 4,4′-bipyridines. Centre: smaller alternatives derived from dimer-forming N-propargylated cyclic hexapeptides (**CP1**, grey) through Sonogashira cross-couplings with Iodopyridines or copper-catalysed azide-alkyne cycloaddition (CuAAC) that have been used as ion transporters[27]. Bottom: cartoon model of supramolecular capsule derived from **CP1** and a tris-azide derivative described in this work.

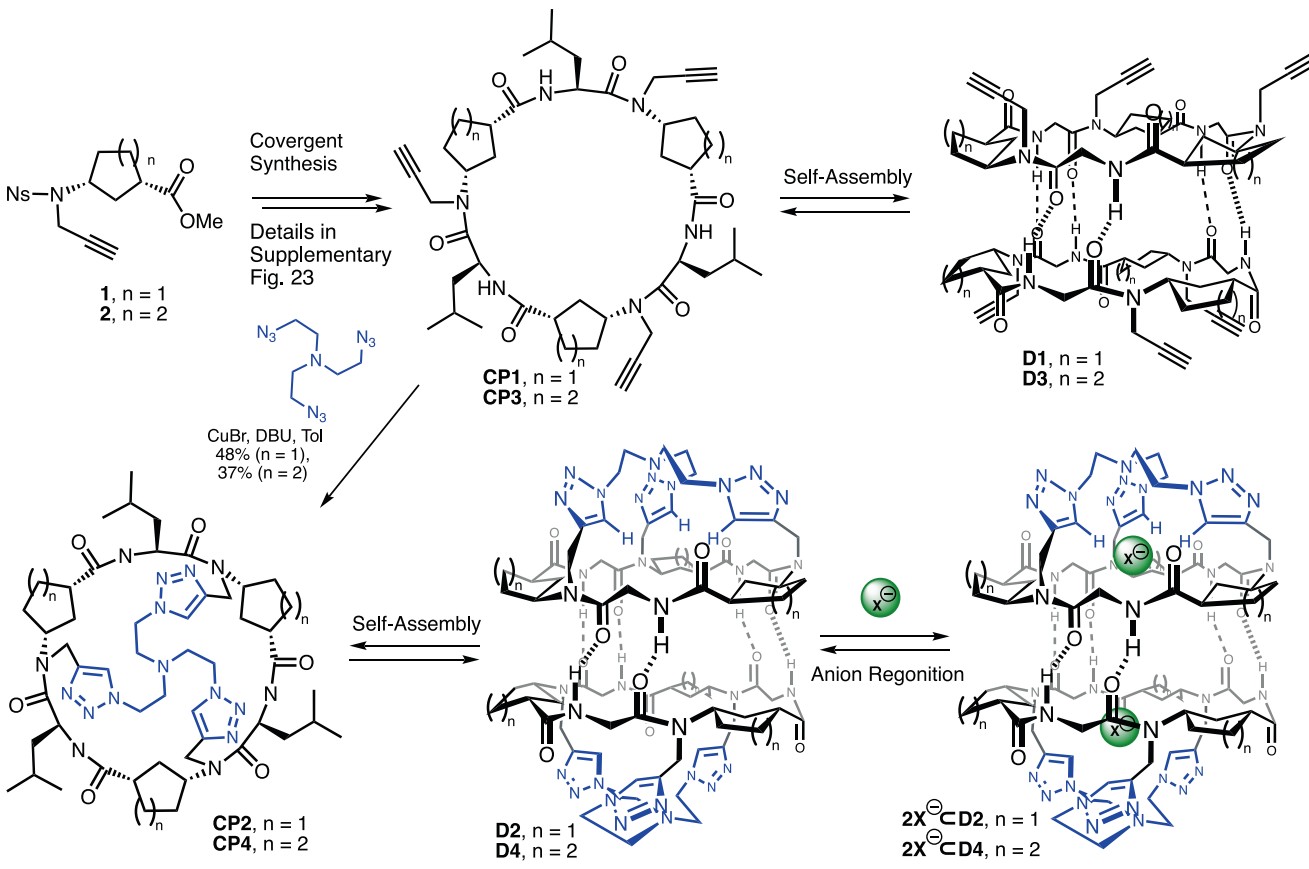

**Fig. 2 | Synthesis and proposed binding model of supramolecular receptor used in this work.** Synthetic strategy used for the preparation of capsules **D2** and **D4** and initially proposed encapsulation model for the recognition of anions (**2X⊖⊂D2** and **2X⊖⊂D4**). Ns denotes Nosyl group (2-nitrobenzenesulfonic derivative).

reaction using the previously optimised method in which catalytic amounts of air-sensitive copper complex [Cu(MeCN)$_4$PF$_6$] (10 mol% per alkyne) were used in the presence of DIEA and TBTA ligand[32,39]. Under these conditions the expected cavitand **CP2** was obtained, although in low yield (15%). After screening for improving reaction conditions, a new protocol was found using copper bromide and DBU in toluene, providing **CP2** in 48% yield[40].

This peptide retained the self-assembling properties of its precursor, **CP1**, adopting the proposed flat conformation and forming the C3-symmetric dimer (**D2**), as denoted by NMR experiments (Supplementary Fig. 1)[41,42]. Clear evidence of dimer formation was derived from the down-field shift of the amide proton (8.04 ppm) whose chemical shift did not change with CP dilution up to 200 μM in CD$_2$Cl$_2$, confirming the large dimerization constant (Ka > 10$^4$ M$^{-1}$) similar to the non-capped derivatives[29]. Further evidence of the dimeric structure derived from the MS (2045.25 [M$_{D2}$ + Na]$^+$, Supplementary Fig. 2) and FTIR spectroscopy, with amide A band a 3286 cm$^{-1}$ and carbonyl vibrations at 1625 and 1529 cm$^{-1}$, characteristic of the antiparallel-type β-sheet structure (Supplementary Fig. 3)[43–45]. The final evidence was obtained from the X-ray crystallography analysis of a crystal obtained from a solution of **D2** in 10% CD$_3$CN in CD$_2$Cl$_2$ from which a dimeric supramolecular capsule held by six hydrogen bonds between both subunits was found (Fig. 3a, b). Interestingly one acetonitrile molecule is entrapped in the inner cavity of this dimer, with its nitrile group pointing towards the protons of the triazolyl moieties of one of the subunits. On the other side, one of the triazole moieties is rotated, arranging two of the nitrogen atoms of the heterocycle towards the internal cavity, perhaps with the goal of partially filling it due to the lack of a suitable guest. We checked if bis-nitrile derivatives, i.e., malononitrile (MN), could interact simultaneously with both tris-triazole motives improving its encapsulation (Supplementary Figs. 4–6). We

found out that it was necessary to add up to 30 equivalents of malononitrile (Supplementary Fig. 5) to a dichloromethane (CD$_2$Cl$_2$) solution of **D2** to get a new derivative (**MN⊂D2**), in a 4 to 1 ratio with respect to the empty capsule. This species was assigned to the encapsulated complex due to the down-field shift of the triazole proton (Δδ = 0.06 ppm). 2D-NMR experiments confirmed that Hβ$_{Acp}$ (*trans*-oriented with carboxy and amino groups) suffered a up-field shift (Supplementary Fig. 6), which is consistent with the incorporation of a malononitrile molecule in the cavity. The weak interaction is most likely due to the 109° angle between the two nitriles, which possibly hinders the simultaneous formation of strong hydrogen bonds with both tris(triazolylethyl)amine moieties. Succinonitrile (SN) was also evaluated, considering its additional methylene moiety allows both cyano groups to be arranged at a 180° angle, although we were concerned that its length would exceed the size of the cavity. Fortunately, the addition of small amounts of SN (~0.5 equiv) over a CD$_2$Cl$_2$ solution of **D2** (6.4 mM) already gave rise to a new set of signals corresponding to **SN⊂D2** (Supplementary Figs. 4 and 7), suggesting a higher affinity than MN. After the addition of 7 equiv, the encapsulated complex (**SN⊂D2**) is already the main component in the mixture, although in a 1.4:1 ratio with **D2** (Supplementary Fig. 7A). Whereas for the MN recognition there is almost no change in the chemical shift of amide protons, for the SN the signal is up-field shifted (7.84 ppm) suggesting a tight binding that reduced the conformational freedom (signals are sharper than those of the free capsule) and stress the hydrogen bond network. The new singlet at 3.30 ppm corresponds to the methylene groups from the entrapped SN molecule as indicated in the 2D-NMR experiments (Supplementary Fig. 7B, C). DFT geometry optimizations confirm the stability of both complexes (Supplementary Fig. 8, see Supplementary Discussion III on page S49 of the Supplementary Information). The main difference between both encapsulated bis-

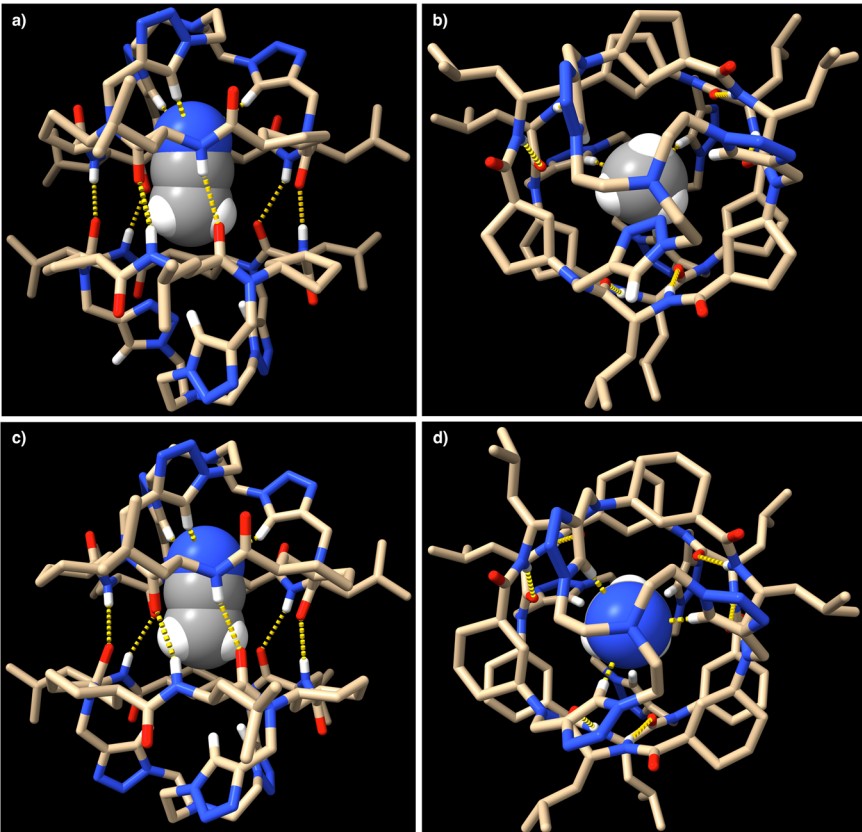

**Fig. 3 | X-ray crystal structures of CP2 and CP4.** Side (**a** and **c**) and top (**b** and **d**) views of the crystal structures of dimeric supramolecular capsules **D2** (top) and **D4** (bottom), respectively. The molecules of acetonitrile entrapped in the cavity are represented in CPK models. The nitrogen of nitrile groups is pointing towards one of the caps close to the triazole protons with a shorter distance in the Ach-based capsule (2.66 Å, bottom) than in the Acp derivative (2.79–2.68 Å, top). Tan colour corresponds to carbon atoms (except for acetonitrile that are represented in grey colour); blue are nitrogen atoms; red are oxygen; white are hydrogen. For clarity, only triazole and amide protons are shown. The yellow dashed lines highlight the hydrogen bonds.

nitrile complexes was the length of hydrogen bonds, which become longer as the size of the encapsulated nitrile increases.

Once the ability of **D2** to form hydrogen-bonded supramolecular inclusion complexes with appropriate guests and the capability of the proton of the tris-triazole cap to form hydrogen bonds with the entrapped molecule was confirmed, the recognition of small anions was evaluated. Initially, we assessed halide recognition using NMR titrations (specifications regarding $^1$H NMR titrations are detailed in Supplementary Discussion I on page S39 of the Supplementary Information). Clearly, the addition of different portions of tetra-butylammonium fluoride to a solution of **CP2** (7.3 mM) in 10% CD$_3$CN/CD$_2$Cl$_2$ gave rise to a new set of signals that were attributed to the recognition of the fluoride anion (Fig. 4). In any case, it was necessary to add ca. 3 equivalents of TBAF to achieve the complete disappearance of the signals corresponding to **D2**. Similar results were obtained with the addition of TBACl (Supplementary Fig. 9) under similar conditions (10% CD$_3$CN/CD$_2$Cl$_2$), but in this case, it was necessary to add more than 15 equivalents of chloride to fully shift the equilibrium towards the new species, suggesting a weaker interaction (Table 1). No changes were observed for the addition of larger halides (bromide and iodide) or nitrate. Careful analysis of the $^1$H NMR spectra of both experiments reveals that, as opposed to what was found in anion recognition with tris(triazolyl) receptors[33–38], the heterocyclic protons are up-field shifted (7.47 ppm). This suggests that these protons must not be participating in halide recognition. On the other hand, the amide proton signals are down-field shifted (from 8.00 to 10.00 ppm for fluoride and to 8.62 ppm for chloride), implying their participation in stronger hydrogen bonds. Conformational changes

derived from the interaction with the anions are clearly evidenced by the splitting of the signal of the methylene linker (singlet at 4.59 ppm for **D2**) between the triazole rings and the cyclic peptide backbone into two doublets at 4.90 and 4.04 ppm. In addition, conformational changes at the CP backbone suggest changes in the flat conformation characteristic of the sheet structure. For example, Hα$_{Leu}$, initially at 4.81 ppm, undergoes an up-field shift to 4.15 ppm for fluoride (4.19 for chloride), while the Hγ$_{Acp}$, initially at 4.73 ppm, undergoes an opposite, albeit milder, down-field shift to 4.90 ppm (4.83 ppm for chloride). Two-dimensional NMR experiments provided evidence of conformational changes, in which nOe cross-peaks that are not present in the empty capsule appear after addition of the halide (Supplementary Fig. 10), such as the one between Hγ$_{Acp}$ and one of the protons of Hβ$_{Leu}$ upon recognition of fluoride. The disappearance, upon complexation, of the strong nOe cross-peaks between H$_{triazole}$ and Hα$_{Leu}$ and of the methylene linker and the Hβ$_{Leu}$ also seems relevant. Further evidence of the halide and CP interactions was found at ESI-MS, in which several ion peaks corresponding to complexes between both components are the main detectable species (Supplementary Fig. 11).

In a search for improving the binding, we discovered that water molecules play a key role in the supramolecular recognition process. The addition of 4 Å molecular sieves, with the aim of drying the solution of TBACl and **D2**, gives rise to the recovery of the starting dimeric capsule. Filtration of the resulting solution and the addition of 10 µL of water provoke the recovery of the chloride-capsule complex (**mCl⁻·nH$_2$O⊂2CP2**) spectrum (Supplementary Fig. 12A). Inspired by this finding, new studies with the other larger and less hygroscopic anions were reviewed to evaluate the role of water molecules in their

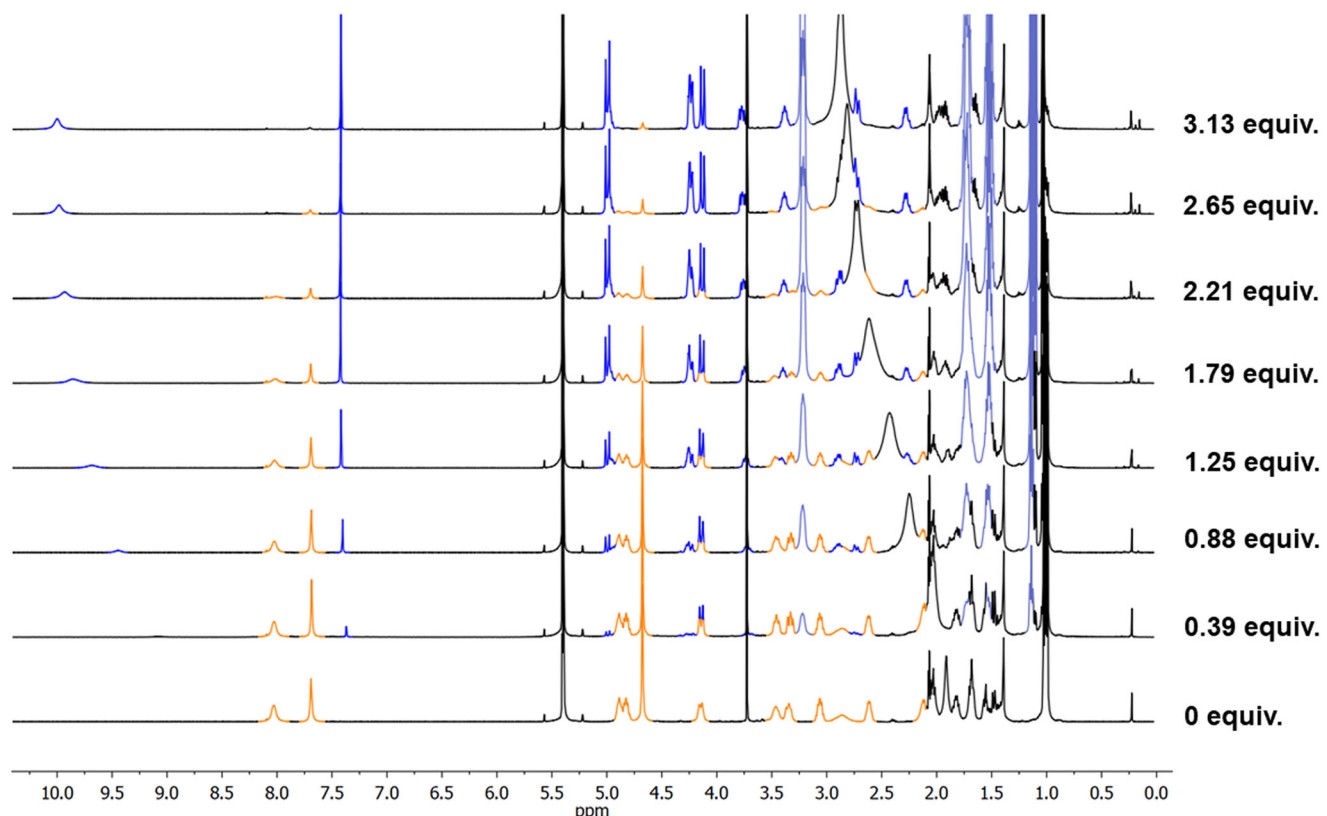

**Fig. 4 | Fluoride binding experiments of CP2.** Stack of NMR spectra of pure **CP2** (bottom) and after the addition of different equivalents of fluoride (TBAF). In blue colour are highlighted the signals corresponding to the new species formed after the addition of the fluoride, light blue denotes the signals corresponding to the tetrabutylammonium counterion. Spectra in 10% CD$_3$CN/CD$_2$Cl$_2$ containing dioxane as internal standard.

recognition. The addition of a few drops of water (7 μL) to the 10% CD$_3$CN/CD$_2$Cl$_2$ solution of **D2** (4.7 mM), resulted in only a small downfield shift of the triazole proton signal, presumably due to water encapsulation (Supplementary Fig. 13); but successive additions of increasing amounts of TBAB yielded similar changes to those already described for chloride additions (28 equivalents added and δ$_{NH}$ 8.37 ppm, Table 1) (Supplementary Fig. 14). Remarkably, even recognition of iodide was also observed, although more equivalents of iodide (86 equiv, Table 1) were required, and the down-field shift of NH signal (8.07 ppm) was even smaller (Supplementary Fig. 15). This definitely underlines the importance of water molecules in the anion

recognition process, most likely due to the ability of the capsule to recognize the hydration spheres of the anions[9]. Next, other anions (see Table 1, Fig. 5, Supplementary Figs. 16–19) were evaluated and a similar behaviour was found; those that were recognised, such as nitrate, acetate or azide ions, only did so in the presence of some water molecules. Larger ions, such as tribromide or hexafluorophosphate did not interact even in the presence of different amounts of water. These results confirm the ability of **D2** to recognise the hydration spheres of different anions. We decided to further study the recognition of acetate to stress out the importance of water molecules in the complexation process by drying this solution. After incubating a solution of **D2** containing 66 equivalents of TBAA with molecular sieves for 48 h, the ¹H NMR signals corresponding to **D2** were recovered, while those assigned to the original complex disappeared. The subsequent addition of a few drops of water (10 μL) immediately re-established the original complex (Supplementary Fig. 12B). Interestingly, while overnight incubation was sufficient to dehydrate the encapsulated chloride complex with molecular sieves, drying the acetate mixture required two days for the full recovery of **D2**, suggesting a tighter binding. Apart from the key role played by water molecules, the chemical shift and number of equivalents of the different anions tested suggest that size, shape, and basicity are important parameters in the recognition process. To confirm the role played by the capsule and the tris(triazolylethyl)amine cap, **CP1** was evaluated for the recognition of anions. TBAF additions over a solution of **D1** in dichloromethane with 10% acetonitrile did not provide any change in its ¹H NMR spectrum, confirming the relevance of the cap in the anion/water cluster recognition (Supplementary Fig. 20). The role of the tetrabutylammonium counterion was also evaluated, for this purpose we decided to make use of the known affinity of crown ethers for alkali cations[46]. The 15-crown-5 is known for being the one whose radius fits better with Na$^+$ [47].

**Table 1 | Key features of the anion encapsulation by tris-triazoyl modified CPs (D2 and D4)**

| Dimer | Anion | Equiv[a] | Molar fraction 3A⁻·nH$_2$O⊂2CP2[b] | NH shift (ppm) | H$_{triazole}$ shift (ppm) | pka |
|---|---|---|---|---|---|---|
| **D2** | F⁻ | 3.13 | 1 | 10 | 7.38 | 3.2 |
| **D2** | Cl⁻ | 24.7 | 0.90 | 8.62 | 7.29 | −8.0 |
| **D2** | Br⁻[c] | 28.1 | 0.92 | 8.35 | 7.28 | −9.0 |
| **D2** | I⁻[c] | 86 | 0.71 | 8.07 | 7.27 | −10.0 |
| **D2** | N$_3$⁻ | 35 | 0.67 | 8.55 | 7.29 | 4.75 |
| **D2** | OAc⁻ | 66 | 0.88 | 8.97 | 7.27 | 4.7 |
| **D2** | NO$_3$⁻[c] | 60 | 0.77 | 8.19 | 7.30 | −1.4 |
| **D4** | F⁻ | 8 | 0 | - | - | 3.2 |
| **D4** | Cl⁻ | 36 | 0 | - | - | −8.0 |

[a]equivalent numbers are given with respect to **CP2** concentration, [b]Molar fraction was calculated at the mentioned maximum number of equivalents of the corresponding anions, [c]7 μL of water were added before the additions of anion solution.

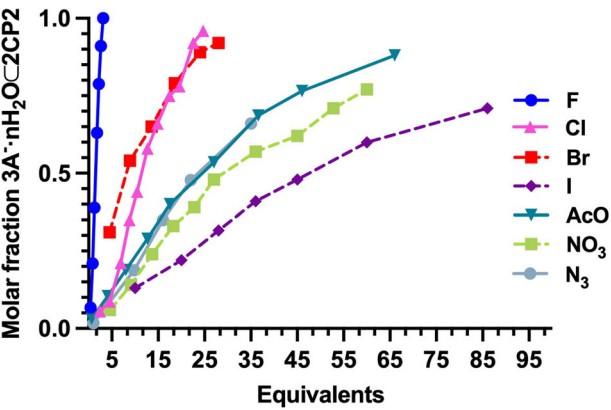

**Fig. 5 | Relative anion binding affinities of CP2.** Molar ratio between **CP2** and the corresponding anion complex represented versus the number of anion equivalent added using data derived from the NMR titration experiments, see Table 1. The dashed lines are used to indicate the anions in which extra water was added to facilitate complex formation.

Therefore, we were able to solubilize NaOAc in deuterated acetonitrile using this crown ether and this solution was used in new titration experiments with **D2** in dichloromethane. Similar results were obtained, although larger amounts of the sodium acetate/crown ether acetonitrile solution were required to shift the equilibrium, most likely due to the lower solubility of the complex under these conditions (Supplementary Fig. 21A). In fact, the NMR spectrum of the mixture two weeks after titration showed a clear increase of the acetate complex with respect to the initial conditions, going from 1.4:1 ratio to 3:1 (Supplementary Fig. 21B). This can be attributed to the slow solubilization of the acetate salt triggered by the formation of the complex. ROESY spectra showed a similar cross-peak pattern to those obtained using tetrabutylammonium as a counterion (Supplementary Fig. 22).

The strong down-field shift of amide protons upon anion addition suggests their involvement in its recognition, consequently, further experiments were carried out to understand this behaviour. It is well established that cyclic peptides made by five-membered ring γ-amino acids (Acp) form heterodimeric complexes with CPs containing six-membered ring γ-amino acids, i.e. **CP3**[42,48,49], which are more stable than the corresponding homodimers. Therefore, we decided to evaluate the importance of capsule integrity in cluster recognition. Consequently, **CP3**, which was prepared following a similar strategy to the one used in the preparation of **CP1** but starting from 3-aminocyclohexanecarboxylic acid (Ach, Supplementary Fig. 23), also forms the corresponding dimer **D3** in dichloromethane solution. **CP3** was mixed with a solution of **D2**, and the resulting mixture showed in the NMR spectra the appearance of a new set of signals that were assigned to the supramolecular aggregate **D2–3** (Fig. 6 and Supplementary Fig. 24). Further evidence of the heterodimeric structure derived from MS experiments (Supplementary Fig. 25) in which the 1840.18 ion peak corresponds to the heterodimeric complex. This confirmed, once again, the higher stability of the heterodimeric aggregates because of the better van der Walls fitting between both cycloalkyl rings[50]. The formation of this heterodimer allowed us to evaluate if dimers containing only one tris(triazolylethyl)amine cap (cavitand **D2–3**) could still recognize anions. Interestingly, after the addition of small portions of fluoride or chloride (TBAF or TBACl) on the CD₂Cl₂ solution of heterodimer **D2–3**, simultaneous recovery of homodimer **D3** and the formation of the corresponding complexes of **CP2** with the anions (**3F⁻·nH₂O⊂2CP2** and **3Cl⁻·nH₂O⊂2CP2**) were observed. Three equivalents of fluoride were also required to shift the equilibrium to the homodimeric components. This confirms that the interaction with hydrated anions is even more favourable than that of

the heterodimer and that both capped subunits are necessary in this process.

Cyclic peptide **CP3** was then transformed into the tris(triazolylethyl) amine capped Ach derivative (**CP4**) using similar conditions in 50% yield (Fig. 2). Therefore, the recognition properties of the new Ach-based capsule topped with the tris(triazolylethyl)amine motif were also studied. These cavitand self-dimerized in dichloromethane solution to form **D4**, as denoted by NMR, MS and FTIR (see Sections 7 and 8 on pages S65 and S100, respectively, of the Supplementary Information). Once again, a single crystal suitable for X-ray diffraction of this compound also confirmed the capsule formation that also entraps one acetonitrile molecule in its cavity (Fig. 3c, d). **D4** has all the triazole protons pointing towards the internal cavity making its structure more symmetric than the Acp-based one (Fig. 3a, b), with similar length in all the interpeptide hydrogen bonds. These are generally longer (2.24 Å) than those of **D2**, which range from 1.91 to 2.35 Å (2.10 Å on average), although capsule **D4** is slightly more compact with a shorter distance between the two nitrogen atoms of the tris(triazolylethyl) caps (14.30 Å versus 14.85 Å). In contrast to the recognition properties of **D2**, additions of more than twenty equivalents of TBAF or TBACl do not result in any change in the NMR spectra of **D4** (Supplementary Fig. 26). This indicates that the Ach-based capsule is not capable of recognising anions, most likely due to the greater rigidity of the six-membered ring of this γ-amino acid that prevents the CP from adopting the appropriate conformation for the recognition of such species.

Both cavitands (**CP2** and **CP4**) also assemble into the heterodimer **D2–4** (Fig. 6 and Supplementary Fig. 27A) when an equimolar mixture of both compounds in non-polar solvents (CD₂Cl₂) is prepared. The appearance of new signals in the ¹H NMR spectra that do not correspond to any of the homodimers confirms the formation of **D2–4**. For example, the broad signal at 4.95 ppm belonging to the Hα_Leu and the one at 4.22 ppm corresponding to one the methylene of the tris(triazolylethyl)amine cap are signals that belong to **CP2** of the heterodimer (Supplementary Fig. 27B). Moreover, the MS also confirms the formation of **D2–4** (Supplementary Fig. 28). Once again, addition of fluoride (TBAF) to this dichloromethane (CD₂Cl₂) mixture prompts the splitting of the heterodimer **D2–4** into the corresponding homodimer **D4** and the complex of hydrated fluoride with cavitand **CP2** (**3F⁻·nH₂O⊂2CP2**). In all cases, the addition of more than three equivalents of fluoride was necessary to achieve complete dissociation of the heterodimers. DFT geometry optimization of both heterodimeric species, **D2–3** and **D2–4**, are shown in Supplementary Fig. 29, which once again confirms the stability of the mentioned dimers and provides further information about the hydrogen bonds length and angles (see Supplementary Discussion III on page S49 of the Supplementary Information).

Crystals suitable for X-ray diffraction were obtained from solutions containing the supramolecular capsules **D2** and the tetrabutylammonium halides (chloride and fluoride, Fig. 6). To our delight, in both cases, the cyclic peptide capsules entrap a new kind of hydrated halide clusters made by three ions, as confirmed by the number of tetrabutylammonium ions that co-crystalized with the peptide capsule. Four and eight water molecules, for chloride and fluoride, respectively, are forming these clusters, confirming the higher tendency of the latter to have larger hydration shells. In both cases, the three halide ions are distributed into six equivalent chemical positions that are shared with another three water molecules. Although for the chloride crystal, all the ion positions form a hexagonal structure with all the positions placed at the same layer, for the fluoride cluster, the six positions are placed at two different levels forming two triangular structures that are 60° rotated with respect to each other. To entrap these clusters, cyclic peptide dimers dissociate to allow amide protons to hydrogen bond with halide ions and water molecules[51]. It is notorious that in the solid state, all the carbonyls are oriented towards the opposite side in which the interaction with the

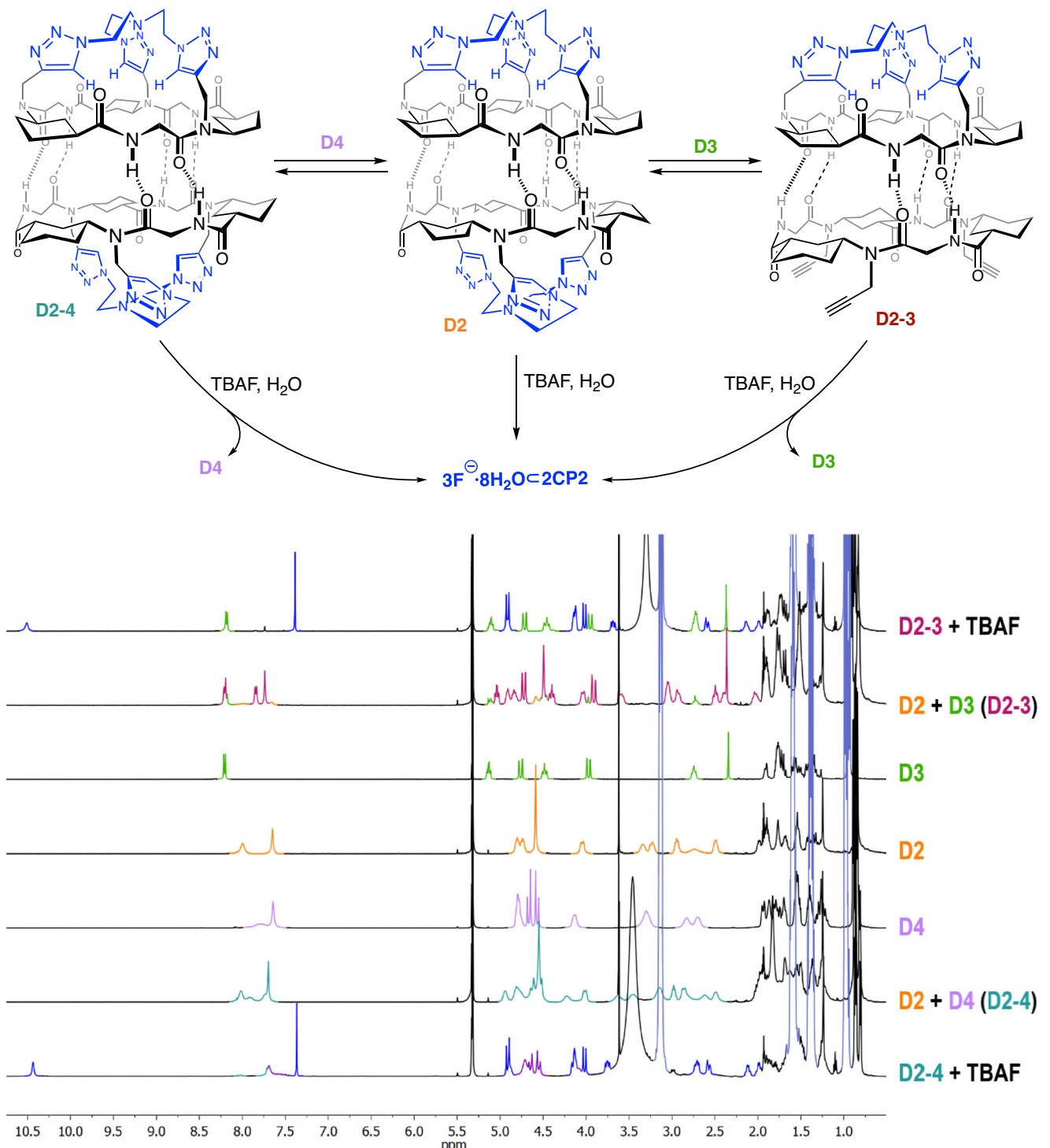

**Fig. 6 | Heterodimeric models and NMR experiments.** Schematic representation of anion recognition processes using heterodimers **D2–3** and **D2–4**, top view. Bottom NMR spectra corresponding to these studies in which the characteristic signals of each component are highlighted with specific colours; orange, green and lavender for homodimers **D2, D3** and **D4**, respectively, dark blue for **CP2** interacting with fluoride, and plum and teal green for heterodimers **D2–3** and **D2–4**, respectively. Spectra in 10% $CD_3CN/CD_2Cl_2$ containing dioxane as internal standard.

anionic cluster occurs, which could explain the observed up-field shift of $H\alpha_{Leu}$ in the NMR spectrum. This conformational change is due to the geometrical variations of the α-amino acids that go from the characteristic β-sheet conformation of flat disc-shaped CPs to a turn-like structure. Interpeptide distance is slightly larger for the encapsulated fluoride cluster than that for chloride despite the larger size of the latter.

With respect to the fluoride-capsule complex (Fig. 7a, c), the unit cell has two non-equivalent complexes (**3F⁻·8H₂O⊂2CP2**). In each complex, in addition to the three water molecules exchangeable by fluoride ions and hydrogen-bonded to the amide proton, there are five other crystallographic positions preferably occupied by water molecules, although fluoride could also partially occupy any of these positions, since it is not possible to unambiguously differentiate both

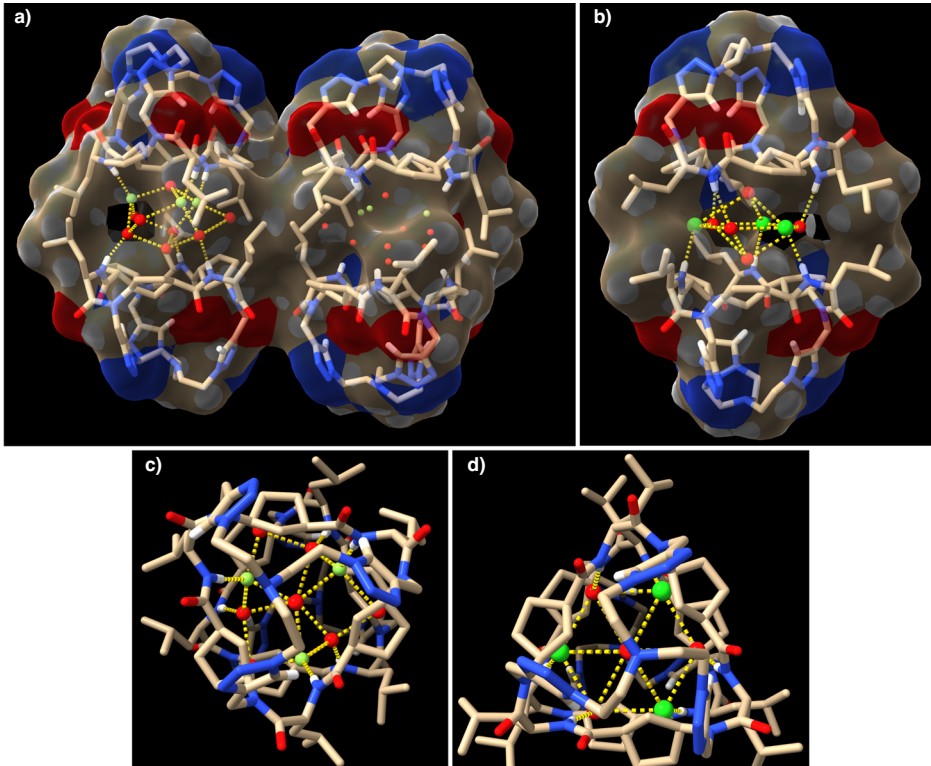

**Fig. 7 | X-ray crystal structures of complexes of CP2 with fluoride and chloride.** Side (**a**) and top (**c**) view of the crystal structure of water-fluoride cluster entrapped in supramolecular capsule **D2** (**3F⁻·8H₂O⊂2CP2**) as twin complexes. In these complexes, three fluoride ions (light green) and eight water molecules are hydrogen-bonded to the amide protons of two cyclic peptides at different planes (the hydrogen bond network (yellow dashed lines) in the cluster is only shown for one of the complexes). Side (**b**) and top (**d**) view of the crystal structure of encapsulated chloride-water cluster between two **CP2** (**3Cl⁻·4H₂O⊂2CP2**). The three chloride ions and water molecules are occupying six equivalent chemical (and crystallographic) positions forming a hexagonal structure, where the two subunits are aligned forming a trigonal bipyramid-shaped (**d**) capsule. Tan colour correspond to carbon atoms; blue are nitrogen; red are oxygen; white are hydrogen; green balls are chloride atoms; light green balls are fluoride.

atoms due to their similar electron densities. In any case, to fulfil the hydrogen acceptor capability of the fluoride ion, we assume that these ions must be located in the positions in which they are bonded to the amide protons of the same cavitand and surrounded by three water molecules forming the first hydration shell of each fluoride (Supplementary Fig. 30). The fluoride occupancy in the two **3F⁻·8H₂O⊂2CP2** complex of each unit cell is not exactly the same (Supplementary Fig. 31), even though both complexes present analogue disposition. With respect to the rest of the water molecules, there are two that are axially placed, forming hydrogen bonds with the fluoride-water exchangeable positions, while the other three are in the cluster equatorial perimeter forming hydrogen bonds with the exchangeable fluoride-water positions (Supplementary Fig. 32). These water molecules are placed at the position in which capsule is not fully closed forming a window. The Leu side chains face each other to create a hydrophobic oval-shaped aggregate that entraps the halide cluster in a non-polar environment.

Concerning the chloride complex (Fig. 7b, d) the six equivalent positions are in the same plane and each chloride ion is hydrogen-bonded to one amide proton with N···Cl distance of 3.21 Å. In this case, the coating with the leucine side chains is less compact, leaving a wider window as compared to the fluoride complex. Furthermore, electron density can only be attributed to a maximum of four water molecules, one of them located at 50% occupancy at the top and bottom of the hexagonal bipyramid and more deeply buried in the cavity of the supramolecular capsule, remaining partially hydrogen-bonded to the three chloride ions. In this crystal structure, the complex of the capsule with the chloride cluster is co-crystallized with the dimer **D3**, forming columns in which **D3** is alternated with the encapsulated trichloride

cluster (**3Cl⁻·4H₂O⊂2CP2**). Within **D3** there is a dioxane molecule occupying three equivalent positions around the ternary symmetry axis of the dimer (Supplementary Fig. 33).

Unfortunately, it was not possible to obtain crystal structures of the complexes formed with the other anions (bromide, acetate, azide, iodide and so on) that would allow unequivocal confirmation of the formation of similar structures with these ions. In any case, the previous characterizations suggest the formation of clusters that are embedded in the equatorial cleft generated by the two CP subunits. To confirm this further, we carried out a detailed analysis comparing the NMR data of the different complexes and the X-ray diffraction data (see Supplementary Discussion II on page S40 of the Supplementary Information), from which we concluded that the recognition process most likely involves the trapping by two **CP2** units of clusters composed of three anions, although we do not have conclusive evidence of such stoichiometry, surrounded by several water molecules depending on the type of anion and its solvation. To this complex we have used the coding **mA⁻·nH₂O⊂2CP2** for the complex with the anion cluster in which $m$ represents the number of anions entrapped between both CPs (most likely three), $A^-$ indicates the type of anion and $n$ represents the number of water molecules that form each anion cluster.

After confirmation of the anion recognition ability of the supramolecular capsule **D2**, transmembrane transport experiments were carried out[1]. For that purpose, lucigenin-trapped liposomes (**LG⊂LUV**) were prepared with which the intravesicular delivery of chloride ions was clearly established (Figs. 8 and 9A)[52]. The transport is slow compared with previously published chloride transporters and a high concentration of capsule is required (EC₅₀~100 μM). Additionally, we

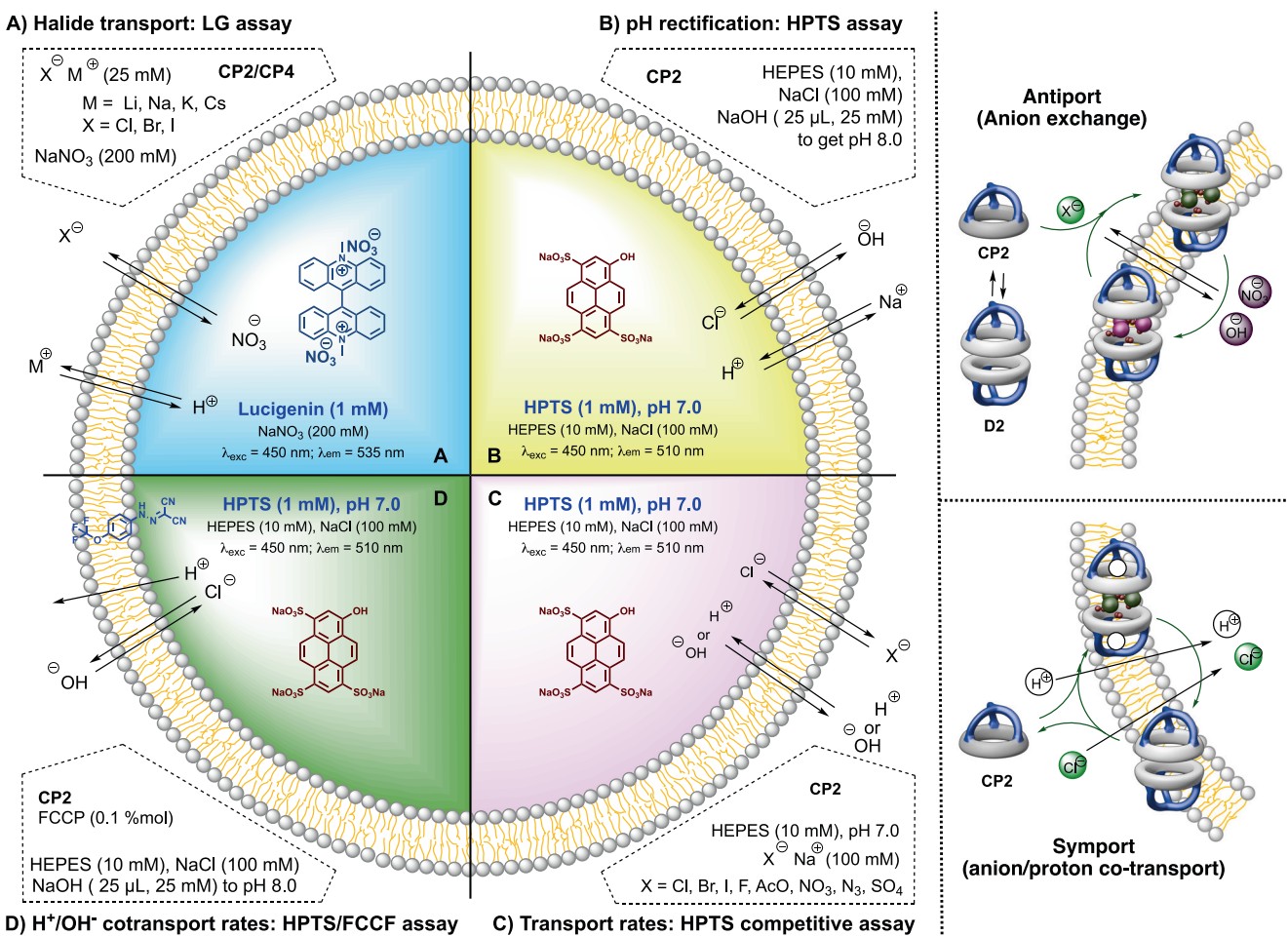

**Fig. 8 | Schematic illustration of anion transport studies.** The transport assays were carried out with liposomes containing different fluorophores to assess the transport mechanism. **A** lucigenin to determine transmembrane halide transport (influx); **B** rectification assay using HPTS at different pH; **C** relative transport rates determination using HPTS; **D** evaluation of the influence of proton transport on the rate-determining step in anion transport using HPTS in the presence of FCCP. On the right is the proposed antiport and symport mechanism. Lucigenin = *N,N* '-Dimethyl-9,9'-biacridinium dinitrate and HPTS = 8-hydroxy-1,3,6-pyrene-trisulfonate; FCCP = Carbonyl cyanide 4-(trifluoromethoxy)phenylhydrazone.

decided to examine the chloride transport ability of **D4** and **D1** using the lucigenin assay. As we expected from the previous NMR findings, neither **D1** nor **D4** were capable of transporting chloride (Supplementary Fig. 34), confirming that the recognition of the ions was necessary to be able to mediate its transport. Apart from that, further variations of the **D2** lucigenin assay, revealed that the transport efficiency did not change with the counterion used (Na[+], Li[+], K[+] or Cs[+]), suggesting that cation is not involved in the transport process (Figs. 8 and 9A3). To confirm the potential antiport transport of nitrate, experiments in which nitrate was substituted for the more hydrophilic sulphate, whose transmembrane transport is more difficult, were carried out (Supplementary Fig. 35)[53,54]. These experiments did not show any significant reduction in chloride transport rates, suggesting that chloride/nitrate exchange must not be involved in the transport mechanism. Therefore, the symport (H[+]/Cl[−]) or antiport (OH[−]/Cl[−]) must be associated with this migration. To confirm the association of chloride transport with change in the pH, HPTS-loaded vesicles (**HPTS⊂LUV**) were used (Figs. 8 and 9B)[55]. Thus, vesicle basification promoted by **D2**, denoted by a fluorescence increase, would be associated with the co-transport of chloride. Unambiguous and fast enhancement of dye emission was found upon creating a pH gradient of almost one unit after the addition of a sodium hydroxide solution to the extravesicular medium, yielding an enhanced activity with EC$_{50}$ = 3 μM, suggesting that anion transport was the rate-limiting step

and not the H[+] or OH[−] co-transport. To clarify this finding, HPTS studies in the presence of a proton transporter (FCCP) were carried out (Figs. 8 and 9D)[56]. As expected no changes in transport rates were found, confirming proton transport was not the limiting step. DLS measurements were then performed, which confirmed both the homogeneity and integrity of the vesicles throughout these experiments and, consequently, the observed fluorescence changes are not due to bleaching or membrane disruption (Supplementary Fig. 36).

Finally, to evaluate the relative rates in anion transport, competitive experiments were carried out using **HPTS⊂LUV** (Figs. 8 and 9C)[52,57]. For this purpose, vesicles whose internal buffer contained sodium chloride (100 mM) at pH 7 were placed in a variety of isosmotic buffer solutions with different sodium salts with other counterions. In this type of experiments, a pH gradient is generated if the transporter (**D2**) facilitates dominant ion influx or efflux depending on anion selectivity. These permeability differences give rise to a membrane potential that drives net proton transport. Vesicle acidification occurs when ion influx is faster than chloride efflux, while ions that are transported more slowly than chloride cause an increase in intravesicular pH. We found that acetate, fluoride, and azide provided vesicle basification, while bromide and iodide were transported faster than chloride, being acetate the slower influxed anion and iodide the faster one. Therefore, **D2** showed Hofmeister pattern (I[−] > Br[−] > NO$_3^-$ ~ Cl[−] > N$_3^-$ > F[−] > OAc[−]) with the exception of nitrate that is almost as fast

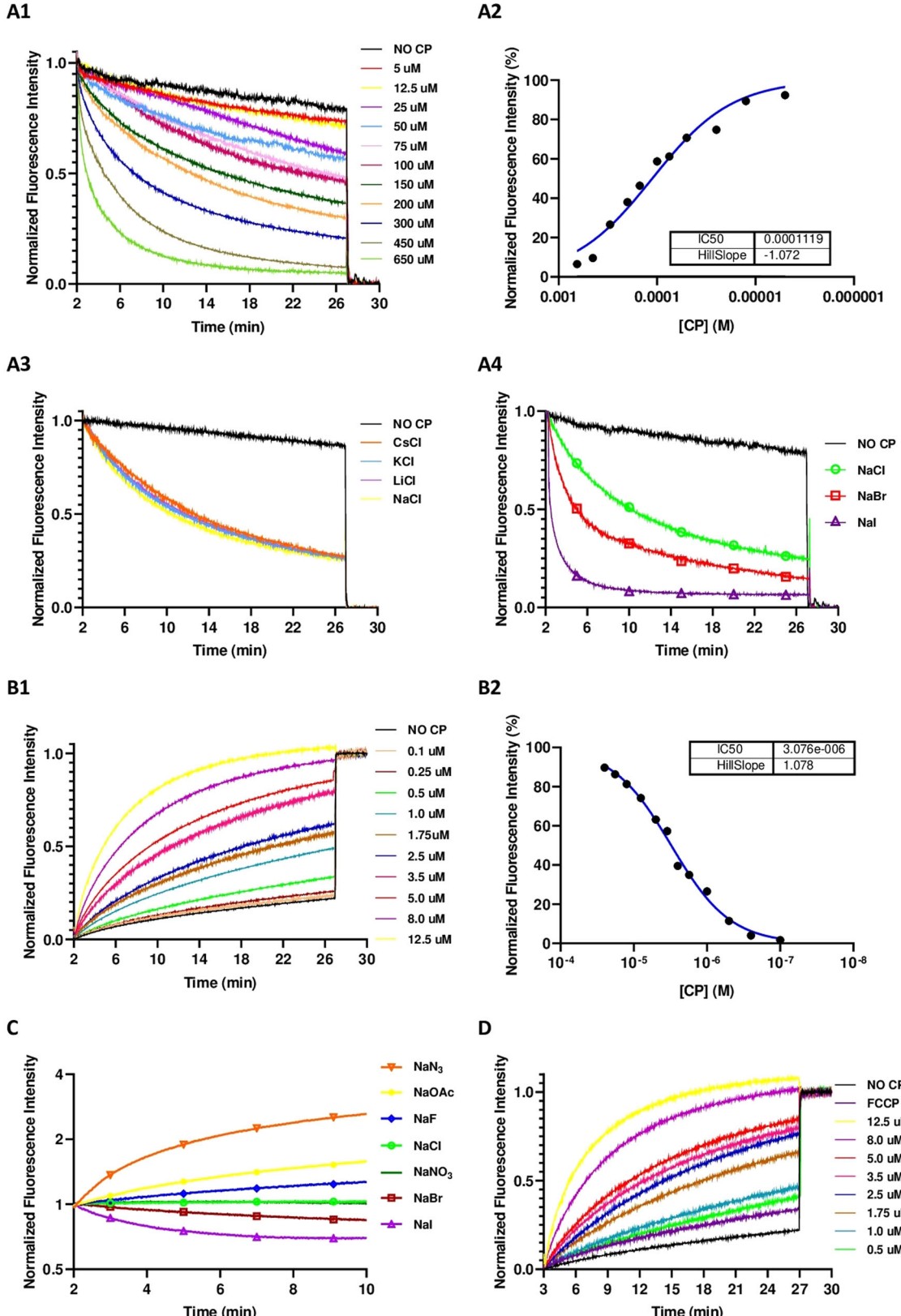

as chloride[58]. The strongly hydrated, salting-out ions, are those with slower transport rates while previous ion recognition experiments showed the need of water molecules in the recognition of anion cluster and the strong binding for fluoride or acetate compared with iodide or bromide. Therefore, it seems that ions exchange ($K_{on}/K_{off}$) at the interfaces must play an important role in the transport process and not

so much the selectivity of the binding itself. All the details regarding the transport assays are carefully addressed in Supplementary Discussion IV on page S53 of the Supplementary Information.

In conclusion, we have presented the design, synthesis, and anion recognition properties of a cyclic α,γ-hexapeptide equipped with a tris(triazolylethyl)amine cap. We have shown that, in presence of some

**Fig. 9 | Anion transport studies to determine transport mechanism.** Plots of transmembrane transport studies with Lucigenin⊂LUV (**A1-A4**) and HPTS⊂LUV (**B1-D**). **A1** Halide influx following Lucigenin emission quenching upon the addition of **CP2** (5–650 µM), λexc = 450 nm and λem = 535 nm. **A2** Lucigenin assay fitting to Hill equation, representing normalized fluorescence intensity at $t$ = 26 min vs **CP2** concentration. **A3** and **A4** describe lucigenin assay studying cation and anion variations, respectively. Different chloride (**A3**) or sodium (**A4**) salts (2.5 µM) were added to the extravesicular media employing a constant **CP2** concentration (75 µM). **B1** HPTS emission enhancement upon the addition of **CP2** (0.1–12.5 µM), λexc = 450 nm and λem = 510 nm. **B2** HPTS assay fitting to Hill equation,

representing normalized fluorescence intensity at $t$ = 26 min vs **CP2** concentration. **C** HPTS assay competition variation. Different sodium salts are contained in the extravesicular media (100 mM), while NaCl (100 mM) remains in the intravesicular one for all the cases. The changes in the HPTS emission are due to the different anions transport rate, cause no base pulse was added. **D** HPTS assay with FCCP addition. The absence of changes in the transport activity under the presence of an efficient proton transporter such FCCP indicates that H+ exchange is not the rate-limiting step. More details about the transport experiments can be found in the "Methods" section of the manuscript and in Supplementary Discussion IV.

anions, the self-assembled CP is able to reaccommodate its backbone conformation from a self-dimerizing flat circular-shaped structure to a triangular one that creates a binding pocket suitable for hosting hydrated anions. Moreover, we have rigorously identified and studied the different aspects involved in the recognition. First, the essential role played by the water molecules that are involved not only in the formation of anion-water clusters, forming the hydration shell of the different anions, and helping them to take the appropriate size and shape, but also in the interaction and entrapment inside the cavity formed by the two peptide hemicapsules. In addition, two key structural components of the cyclic peptide were identified: the five-member ring γ-amino acid and the tris(triazolylethyl)amine cap, without which the anion accommodation is not possible. Furthermore, we have shown that the cationic counterion does not play any relevant role in the supramolecular recognition, which supports our theory of the three main actors: capped peptide, water and anion.

Finally, the anion transporting properties were also explored. Our studies have shown that **D2** can transport different anions across model lipid membranes, with a tendency to favour the transport of anions with weaker hydration spheres, which are generally more weakly recognized. Transport is associated with the modification the pH of the intravesicular medium, most likely through a Cl−/H+ symport mechanism, although the Cl−/OH− antiport exchange could not be ruled out, the rate-limiting step being the migration of the anions. The peptide composition, design simplicity and anion transporting properties linked to pH regulation activity open up several avenues for the therapeutic use of these self-assembled systems[59].

## Methods

### General ¹H NMR titration protocol
Stock solutions of internal standard, either dioxane (2.5 mM) or TMSS (0.15 mM), in a mixture of CD₃CN in CD₂Cl₂ (10% v/v) were prepared. These solutions were used to prepare all the samples needed in the titration experiments, in order to keep the concentration of the internal standard constant throughout the experiments. The signals from both standards, dioxane (s, 3.6 ppm in CD₂Cl₂) or TMSS (s, 0.18 ppm in CD₂Cl₂), were used to calculate the concentration of all the components along the titration experiments.

Generally, 450 µL of the solution containing the corresponding peptide (**CP2**, **CP3** or **CP4**, ca. 5 mM) was placed in the NMR tube. After recording the ¹H NMR spectra of the starting sample, successive additions of the corresponding titrant were made via micropipette and the corresponding spectrum was taken.

### General procedure for vesicle preparation
Vesicles were prepared, under argon, in a round bottom flask by slow evaporation of a solution of EYPC in CHCl₃ (25 mg/mL, 1 mL) to form a thin and homogeneous film on the flask surface. The film was dried overnight under high vacuum and then carefully hydrated with the intravesicular aqueous media. The resulting mixture was tumbled for 1 h in the rotavapor at 180 r.p.m. but at atmospheric pressure. Every 15 min, the rotavapor rotation angle was changed (60°, 50°, 45° and 35°) in order to get and homogeneous dispersion. After that, the milky sample was subjected to 11 freeze-thaw cycles (N₂ (l) → 40 °C water),

and the resulting suspension was extruded (25 times) across polycarbonate membrane (200 nm pore size). Finally, the suspension was passed through a size exclusion column (Sephadex G-25) previously equilibrated with the isosmotic extravesicular medium[60]. The resulting vesicle suspension was taken up in a total volume of 5 mL, giving an approximate lipid concentration 6.6 mM. Size and particle number consistency between different vesicles batches were checked using DLS.

### General procedure for transport measurement
In a plastic cuvette containing a 4 mm diameter stirring bar, the previously prepared vesicle suspensions (50 µL) were dispersed in the extravesicular media (1950 µL). The cuvette was placed with moderate stirring in the fluorometer equipped with a module that allows the successive additions of titrant during the measurement in the dark. Data (fluorescence emission band at 535 nm for lucigenin assays or 510 nm for HPTS experiments) were collected for thirty minutes every second. One minute after the experiment started, aqueous solutions of NaCl (25 µL, 2 M, lucigenin assay) or NaOH (25 µL, 0.5 M, HPTS assay) were added. After an additional minute (minute 2 of the measurement), a solution of the CP at different concentrations (25 µL, in iPrOH for the lucigenin assay or in DMSO for the HPTS) were added. Finally, after 27 min, an aqueous solution of Triton (50 µL, 10% v/v) was added to provoke liposome lysis. After three more minutes of stabilization, the resulting signal was used for the data normalization.

### DLS measurements
For each batch of vesicle prepared, some control measurements were performed before the transport experiments to check the homogeneity between batches. The size, polydispersity index and particle concentration were measured using the same volumes as in the fluorescence assay. Four samples were collected and checked for each essay. Regarding the lucigenin experiments (Supplementary Fig. 36), the first sample, containing only the vesicles dispersed in the extravesicular media (blue line), the second sample, which corresponds to the mixture after the addition of the aqueous solution of NaCl (green line), the third measurement was carried out after the incorporation of CP (pink line), and finally, the fourth sample was done after the addition of Triton X-100 (orange line). The four samples were stirred for 25 min, and after a 5 min lag resting period, the DLS were recorded. For the HPTS assay (See Supplementary Fig. 36) in HEPES buffer (10 mM, NaCl 100 mM, pH 7), the analyzed mixtures correspond to the initial conditions (blue line), and after successive additions of NaOH (green line), cyclic peptide (pink line) and Triton (orange line).

Notice that samples measurement indicates monodispersity, in view of the sharp slope of the correlation curves, which was maintained in all samples except after the addition of the surfactant, Triton X-100, when the vesicles undergo lysis. After lysis, the faster decay of the correlation coefficient is a signal of the smaller particle size. Also, the slope, less sharp, indicates higher polydispersity. We measured the particle concentration, which also indicated homogeneity between batches and confirmed the stability of the vesicles until their lysis.

## DFT geometry optimizations

Computational studies were carried out using the sources from Centro de Supercomputación de Galicia (CESGA). All calculations were performed with the Gaussian 16 rev. C01 package. The geometries used in the calculations were based on the crystal structures derived from this study. Calculations were performed in vacuum. We carried out DFT geometry optimizations using B3LYP with the moderate-size basis set 6-31G (d, p). We also included GD3BJ as empirical dispersion.

## Data availability

Supplementary Information: Detailed descriptions of the synthesis and characterization of key compounds, including NMR spectra ($^1$H and $^{13}$C, NOESY and/or ROESY) and FTIR spectra of peptides **CP2**, **CP3** and **CP4** are provided. DFT structure coordinates are provided in a separate Excel file as Source Data. Crystallographic data for the structures reported in this Article have been deposited at the Cambridge Crystallographic Data Centre, under deposition numbers CCDC-2311116 to CCDC-2311119. Copies of the data can be obtained free of charge from The Cambridge Crystallographic Data Centre via https://www.ccdc.cam.ac.uk/structures. All data is available from the authors on request. Source data are provided with this paper.

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

## Acknowledgements

This work was supported by the Spanish Agencia Estatal de Investigación (AEI) (PID2019-111126RB-I00 and PID2022-142440NB-I00), the Xunta de Galicia (ED431C 2021/21 and Centro de investigación do Sistema universitario de Galicia accreditation 2023-2027, ED431G 2023/03), and the European Union (European Regional Development Fund - ERDF). We also thank the ORFEO-CINCA network and Mineco (RED2022-134287-T). V.L.-C. thanks the Xunta de Galicia for her research contract (ED481A-2019/117). All calculations were carried out at the Centro de Supercomputación de Galicia (CESGA).

## Author contributions

V.L.-C., A.F. and J.R.G. conceived the concept and designed the experiments. A.F. carried out the first synthesis of **CP1** and **CP2**. V.L.-C. performed all NMR and fluorescence experiments and cyclic peptide synthesis. V.L.-C., M.A. and J.R.G. analysed data. A.L.L.-S. conducted and carried out X-Ray data collecting and analysis. V.L.-C. performed the DFT simulations. M.A. and J.R.G. supervised the project. V.L.-C. and J.R.G. wrote the manuscript with contributions from all authors.

## Competing interests

The authors declare no competing interests.
