## [Peer Review File · Nature Communications]

REVIEWER COMMENTS

Reviewer #1 (Remarks to the Author):

This manuscript describes the synthesis and characterization of a supramolecular dimer which can bind anions. The manuscript is well-written and the methods used to characterize the compounds (including X-ray diffraction, NMR, transmembrane transport experiments) are fairly thorough. The results showing how the macromolecules bind the anions, and the solvating water molecules is interesting and relevant to the larger topic of selective binding of ions. I would recommend this paper for publication without revision.

Reviewer #2 (Remarks to the Author):

Granja and coworkers demonstrated supramolecular dimerization of a capped cup-shaped cyclic peptide and its ability to recognize different anions in the form of hydrated clusters. The formation of supramolecular dimers or anion binding and transport have been explored by several groups using different systems. What is interesting about the present study is the importance of water in the binding. Although the anion binding is studied in Dichloromethane and Acetonitrile mixture, a trace amount of water present in the solvents/chemicals is essential for binding. The removal of water traces by adding molecular sieves to the host-guest complex resulted in the removal of anion from the guest. The effect is reversible wherein the addition of water again leads to the binding. In supramolecular chemistry, the role of solvents in a study has high significance on the behavior of a molecule/material yet rarely their role is completely understood. Hence the present study focussing on the binding of anion-water clusters is significant.

The study is experimentally rich with clean characterization and analysis, but the presentation can be improved.

Please provide integrations in the supporting figures for the ^1H NMR spectra of the supramolecular dimers and the complexes.

There are multiple typos throughout the manuscript and SI for scientific and non-scientific terms (for example, RMN for NMR), please correct.

The authors demonstrate that the CP1-CP4 exists as the corresponding supramolecular dimeric forms D1-D4 in solution. In the discussion/figures, authors represented the dimers using different notations like CP, 2CP, and D leading to confusion and difficulty in understanding the content (for example, refer to Figure 5 and the corresponding discussion).

The authors conclude that the recognition process most likely involves dimeric capsules and trimeric anions. It was also mentioned that experimental evidence regarding this is provided in the case of halides F⁻ and Cl⁻ using crystal structures and mass but not in the case of other anions (Page 13 of the manuscript). It was unclear how the authors concluded the stoichiometry of binding for the anions other than F⁻ and Cl⁻ without mass, crystal structure, or ITC. There is a significant increase in the size of the anions N₃⁻, OAc⁻, and NO₃⁻ compared to halides. The cavity size does not seem to be able to fit three big anions. If there is no clear confirmation, I suggest mentioning the uncertainty about stoichiometry rather than generalizing that all anions could be trimers. In that case, it is suggestible to modify the title also to “anion-water clusters” instead of “trimeric anionic-water clusters”

Please provide a brief explanation of the results obtained from DFT

TOC can be improved; the current TOC does not represent the anion-water clusters clearly in the supramolecular dimer.

Reviewer #3 (Remarks to the Author):

This paper reports the synthesis of a cyclic peptide which is furnished with a cap to produce a hemispherical unit which dimerises through hydrogen bonding interactions. The two halves of the sphere are shown to open up to encapsulate fluoride and chloride in solution and in the solid state. In the crystal structure, there is clear evidence of water being involved in the formation of a specific cluster. In solution, water is required, but the structural nature of the cluster is not clearly demonstrated experimentally. Membrane transport is demonstrated, and explored using a conventional set of experiments.

My view is that this paper contains some very nice supramolecular chemistry work, and that the importance of water in the process is a highlight, though not as important as the introduction makes out. The experiments and analysis are all ably conducted and meet all the requirements of the field. It's all very well done, and will be likely used as an example of hydrated anions being encapsulated (though plenty of crystal structures exist of anion complex hydrates already) rather than built upon directly by others.

The introduction contains some statements which need work - electrostatic interactions are a noncovalent interaction (top of p2), proteins are anion-selective and usually hold anions in a largely desolvated state. It's too much to call a host-guest complex a 'new state of matter.'

There should be consideration of the possibility of deprotonation by fluoride. Solvent needs to be given clearly in the main text when talking about NMR titrations etc.

28th March 2024

The following pages provide point by point answers to each of the questions, comments or suggestions proposed by the different reviewers.

Reviewer: 1

– *Comment. “This manuscript describes the synthesis and characterization of a supramolecular dimer which can bind anions. The manuscript is well-written, and the methods used to characterize the compounds (including X-ray diffraction, NMR, transmembrane transport experiments) are fairly thorough. The results showing how the macromolecules bind the anions, and the solvating water molecules is interesting and relevant to the larger topic of selective binding of ions. I would recommend this paper for publication without revision.”*

Response. We greatly appreciate the positive comments about our work, how it was carried out and the characterization of the compounds and their derivatives. Thank you.

Reviewer: 2

– *Comment. “Granja and coworkers demonstrated supramolecular dimerization of a capped cup-shaped cyclic peptide and its ability to recognize different anions in the form of hydrated clusters. The formation of supramolecular dimers or anion binding and transport have been explored by several groups using different systems. What is interesting about the present study is the importance of water in the binding. Although the anion binding is studied in Dichloromethane and Acetonitrile mixture, a trace amount of water present in the solvents/chemicals is essential for binding. The removal of water traces by adding molecular sieves to the host-guest complex resulted in the removal of anion from the guest. The effect is reversible wherein the addition of water again leads to the binding. In supramolecular chemistry, the role of solvents in a study has high significance on the behavior of a molecule/material yet rarely their role is completely understood. Hence the present study focussing on the binding of anion-water clusters is significant.*

The study is experimentally rich with clean characterization and analysis, but the presentation can be improved.”

Response. We thank the referee for his/her interest and positive comments on our work and this manuscript. We are very pleased to hear his/her comments about the quality and significance of our work and the careful characterization and analysis of the results that we report here. Following his/her advice, the presentation was revised to improve it.

Juan R. Granja Guillán, Ph.D.

Professor, Department of Organic Chemistry,
Center for Research in Biological Chemistry and Molecular Materials (CIQUS). University of Santiago
Phone +34-881 815746. Fax +34-881 815704. E-mail juanr.granja@usc.es

– *Comment. “Please provide integrations in the supporting figures for the ^1H NMR spectra of the supramolecular dimers and the complexes.”*

Response. We have included all the integrations in the ^1H NMR spectra of the supramolecular dimers and complexes in the supporting information, in the section 8. Thank you very much for your kind suggestion.

– *Comment. “There are multiple typos throughout the manuscript and SI for scientific and non-scientific terms (for example, RMN for NMR), please correct.”*

Response. We are very sorry for the inconvenience related to the quality of the written document. We have edited both the manuscript and SI to correct all the typographical errors in these documents. Non-scientific terms such as RMN were also corrected.

– *Comment. “The authors demonstrate that the **CP1-CP4** exists as the corresponding supramolecular dimeric forms **D1-D4** in solution. In the discussion/figures, authors represented the dimers using different notations like CP, **2CP**, and **D** leading to confusion and difficulty in understanding the content (for example, refer to Figure 5 and the corresponding discussion).”*

Response. Again, we are very sorry for these errors, which cause some confusion and may make difficult to understand the content of the work. In general, we have used the term CP for the abbreviation of Cyclic Peptide generically, while **CP1-CP4** are used to describe the molecular structure, in which all atoms are covalently linked, corresponding to the four cyclic peptides used in this work. The terms **D1-D4** for homodimers or the corresponding heterodimers (**D2-3** or **D2-4**) are intended to emphasize the supramolecular species that are present predominantly in the organic solvents used in this work (dichloromethane or acetonitrile in dichloromethane). Therefore, we preferably use **D_n** to describe the supramolecular species that are mainly present in solution experiments or in the crystals. The same can be said for heterodimeric species, in which two different cyclic peptides are mixed and assembled in solution forming hydrogen bonds between both molecules. Therefore, **D** (dimer) is used to describe the supramolecular species in which both peptide units are linked by hydrogen bonds. To clarify this possible ambiguity, we have modified the nomenclature for complexes with anions, in which the two cyclic peptides are not directly linked to each other by hydrogen bonds, using the new numeration, **mA⁻.nH₂O⊂2CP2**, for the complex with the anion cluster in which “*m*” represents the number of anions entrapped between both CPs (most likely three), “*A*” indicates the type of anion (F^- , Cl^- , Br^- , I^- , AcO^- , N_3^- or NO_3^-) and “*n*” represents the number of water molecules that form the entrapped anion clusters.

– *Comment. “The authors conclude that the recognition process most likely involves dimeric capsules and trimeric anions. It was also mentioned that experimental evidence regarding this is provided in the case of halides F^- and Cl^- using crystal structures and mass but not in the case of other anions (Page 13 of the manuscript). It was unclear how the authors concluded the stoichiometry of binding for the anions other than F^- and Cl^- without mass, crystal structure, or ITC. There is a significant increase in the size of the anions N_3^- , OAc^- , and NO_3^- compared to halides. The cavity size does not seem to be able to fit three big anions. If there is no clear confirmation, I suggest mentioning the*

Juan R. Granja Guillán, Ph.D.

Professor, Department of Organic Chemistry,
Center for Research in Biological Chemistry and Molecular Materials (CIQUS). University of Santiago
Phone +34-881 815746. Fax +34-881 815704. E-mail juanr.granja@usc.es

uncertainty about stoichiometry rather than generalizing that all anions could be trimers. In that case, it is suggestible to modify the title also to "anion-water clusters" instead of "trimeric anionic-water clusters".

Response. We appreciate very much this comment regarding stoichiometry with other anions different to fluoride and chloride. Unfortunately, despite having attempted several experimental techniques, including different crystallization conditions, we were unable to unequivocally confirm the proposed stoichiometry. In any case, we consider that the invariability of the C3 symmetry of the cyclic peptide, as reflected in the NMR spectra, upon the addition of the different anions support our stoichiometry hypothesis. Another clear evidence found, also derived from the NMR studies, comes from the fact that the cyclic peptide adopts, after the additions, very similar conformations to that observed for the complexes with chloride and fluoride ions. The main difference between them is the chemical shift of amide proton (NH), whose down-field shift is correlated with the strength of the hydrogen bonds with the different anions and not with other conformational changes. In any case, following his/her advice we have modified the title of our work and also the abbreviation for each complex (**mA⁻.nH₂O-2CP2**), in which "m" represents the number of anions (A⁻) that form the entrapped clusters.

– Comment. "Please provide a brief explanation of the results obtained from DFT."

Response. We thank this referee for asking about the DFT studies, which mainly confirm the stability of the proposed dimers and give some additional structural insights, hydrogen bonds distances, angles, and so on, to those species that have not crystallized. The truth is that we decided not to go on describing these results, which mainly confirmed what was deciphered with other experiments, so as not to make this article too long. Following the referee demand, we have decided to include an additional section in the supporting information (Supplementary Discussion III) in which we briefly describe the most relevant DFT results.

– Comment. "TOC can be improved; the current TOC does not represent the anion-water clusters clearly in the supramolecular dimer."

Response. We thank the reviewer for his kind suggestion. Therefore, a new TOC figure is attached to better represent the anion-water clusters in the supramolecular dimer.

Reviewer: 3

– Comment. "This paper reports the synthesis of a cyclic peptide which is furnished with a cap to produce a hemispherical unit which dimerises through hydrogen bonding interactions. The two halves of the sphere are shown to open up to encapsulate fluoride and chloride in solution and in the solid state. In the crystal structure, there is clear evidence of water being involved in the formation of a specific cluster. In solution, water is required, but the structural nature of the cluster is not clearly demonstrated experimentally. Membrane transport is demonstrated, and explored using a conventional set of experiments."

Juan R. Granja Guillán, Ph.D.

Professor, Department of Organic Chemistry,
Center for Research in Biological Chemistry and Molecular Materials (CIQUS). University of Santiago
Phone +34-881 815746. Fax +34-881 815704. E-mail juanr.granja@usc.es

My view is that this paper contains some very nice supramolecular chemistry work, and that the importance of water in the process is a highlight, though not as important as the introduction makes out. The experiments and analysis are all ably conducted and meet all the requirements of the field. It's all very well done, and will be likely used as an example of hydrated anions being encapsulated (though plenty of crystal structures exist of anion complex hydrates already) rather than built upon directly by others."

Response. We thank the referee for the positive review of our supramolecular chemistry work, highlighting the important role played by water molecules in the recognition properties of capped α,γ -cyclic peptides. We are glad to find his interest and confirmation about the analysis and experimental planification and development to clearly establish the encapsulation of hydrated clusters. It is true that there are a number of crystal structures having anion complexes hydrated, although only a few of them describe polyanionic cluster of the type here reported.

– Comment. "The introduction contains some statements which need work - electrostatic interactions are a noncovalent interaction (top of p2), proteins are anion-selective and usually hold anions in a largely desolvated state. It's too much to call a host-guest complex a 'new state of matter.'"

Response. We very much regret the lack of precision in the wording of some concepts mentioned in the introduction. We have modified the introduction to include the "directional" noncovalent interactions to differentiate to those pure electrostatic between cations and anions. Now we mention: "The design principles used in **this** development are generally dominated by **directional** noncovalent interactions, such as halogen, chalcogen or hydrogens bonds, among others, rather than by purely electrostatic ones, as they generally provide better selectivity."

It is true that the majority of anion-selective proteins recognize them in their desolvated state, however some authors have proposed that the selective transport of fluoride carried out by the FLUC protein, which presents a remarkable selectivity against chloride (Gomez, D. T.; Pratt, L. R.; Asthagiri, D. N. Rempe, S. B. *Acc. Chem. Res.* 2022, 55, 2201–2212) or the recognition of other anions such as sulfate (K. Balamurugan, M. T. Pisabarro, *ACS Omega* 2021, 6, 25350–25360) could be related to the recognition of their hydrates in a similar way to the selective transport of sodium (versus potassium) in the sodium channel proteins.

We have modified the mentioned sentence at the introduction in the following way:

"In some cases, the ion selectivity in aqueous media does not depend as much on the relative size or charge of the ions, but on the size of their corresponding hydrates.^[9,10] In fact, some authors have proposed that the selectivity in fluoride transport by FLUC protein could be explained in a similar way to the K^+/Na^+ selectivity of sodium ion channels.^[11] Additionally, the greater solubility of negatively-charged proteins may be related to stronger hydration free energies for negatively charged groups on protein surfaces than their cationic counterparts.^[12]"

Regarding the term "new state of matter", we would like to make some clarifications to this concept. From our point of view, molecular capsules are much more complex molecular containers than the "classic" molecular receptors characteristic of host/guest chemistry, since

Juan R. Granja Guillán, Ph.D.

Professor, Department of Organic Chemistry,
Center for Research in Biological Chemistry and Molecular Materials (CIQUS). University of Santiago
Phone +34-881 815746. Fax +34-881 815704. E-mail juanr.granja@usc.es

they can recognize more than one substrate, they can incorporate a variable number of solvent molecules, they have a dynamic behavior in which the entrapped molecules flow at different rates among others. With that phrase we intended to emphasize that molecular capsules, such as those reported by Fujita and other authors, generate closed spaces that are more voluminous than conventional molecular receptors, can incorporate *ad hoc* reactive elements, trap macromolecules or proteins, maintaining or modifying their reactivity in which the substrates pass through their molecular walls, so they can be filtered or sorted on demand among other unique properties. In any case, since it seems that we have not depicted clearly enough this concept, this sentence has been modified to indicate the following: “**They can be considered to create a new kind of molecular environment in which trapped molecule(s) exist.**”

– *Comment.* “*There should be consideration of the possibility of deprotonation by fluoride. Solvent needs to be given clearly in the main text when talking about NMR titrations etc.*”

Response. We appreciate his advice and concerns very much. Deprotonation of amide groups was something we were very worried about from the beginning of this experiment, and this was one of the reasons why we decided to carry out all the experiments in dichloromethane and not in chloroform, in which, in addition to its well-known decomposition forming HCl, the deprotonation of chloroform with fluoride generates DF (deuterium fluoride) that exchange with amide NH and consequently the integration values of this proton signal decreases with time. This is something we have never seen in our studies. We are very confident that deprotonation with fluoride is not taking place in these experiments, as confirmed by the x-ray crystallographic studies. We believe that the recognition of the anion clusters reduces the intrinsic basicity of this anion. Since we have now included the ^1H NMR integrals, this can be checked in the proton spectra corresponding to $3\text{F}\cdot 8\text{H}_2\text{O}\cdot 2\text{CP}2$, in the Supplementary Information, section 8.

Solvent conditions were added in the manuscript following the referee advise. In general, deuteriochloroform was used for the NMR characterization of dimers (homo and heterodimers) and titration experiments with bis-nitrile compounds (malononitrile and succinonitrile). For those experiments in which anion were added, a mixture of acetonitrile and dichloromethane (10%) was used.

Highlighted manuscript is attached to this letter.

Juan R. Granja Guillán, Ph.D.

Professor, Department of Organic Chemistry,
Center for Research in Biological Chemistry and Molecular Materials (CIQUS). University of Santiago
Phone +34-881 815746. Fax +34-881 815704. E-mail juanr.granja@usc.es

Recognition of anion-water clusters by peptide-based supramolecular capsules

Victoria López-Corbalán,[†] Alberto Fuertes,[†] Antonio L. Llamas-Saiz,[‡] Manuel Amorín,[†] & Juan R. Granja^{†*}

[†]*Centro Singular de Investigación en Química Biolóxica e Materiais Moleculares (CIQUS) and Organic Chemistry Department, University of Santiago de Compostela, and [‡]Unidad de Rayos X; Área de infraestructuras de Investigación, RIAIDT Edificio CACTUS, University of Santiago de Compostela, 15782, Santiago de Compostela, (Spain).*

The biological and technological importance of anion-mediated processes has made the development of improved methods for the selective recognition of anions one of the most relevant research topics today. The hydration sphere of anions plays an important role in the functions performed by anions by forming a variety of cluster complexes. Here we describe a supramolecular capsule that recognizes new hydrated anion clusters. These clusters are **most likely composed of three ions **that form** hydrated C₃ symmetry complexes that are entrapped within the supramolecular capsule of the same symmetry. The capsule is made of self-assembled Acp-based cyclic peptide equipped with a tris(triazolylethyl)amine cap. To recognise the hydrated anion clusters, the hexapeptide capsule must disassemble to entrap them between its two subunits.**

Anion recognition is a biologically and technologically relevant process in which *ad hoc* designed receptors are able to bind to a specific ion triggering a specialised function.^[1,2] In the last years, a large number of new synthetic receptors have been developed for this purpose.^[3,4] The design principles used in **this** development are

generally dominated by **directional** noncovalent interactions, such as halogen, chalcogen or hydrogens bonds, among others, rather than by purely electrostatic ones, as they generally provide better selectivity.^[5,6] **In this regard**, nature is able to differentiate **biologically** relevant ions of similar properties, such as fluoride or chloride, to carry out selective cross-membrane transport with remarkable selectivity.^[7,8] The strong hydration of anions is one of the main obstacles to their recognition due to the additional energetic penalty paid for their desolvation during host–guest binding. In **some** cases, the ion selectivity in aqueous media does not depend as much on the relative size or charge of the ions, but on the size of their corresponding hydrates.^[9,10] **In fact, some authors have proposed that the selectivity in fluoride transport by FLUC protein could be explained in a similar way to the K^+/Na^+ selectivity of sodium ion channels.**^[11] Additionally, the greater solubility of negatively-charged proteins may be related to stronger hydration free energies for negatively charged groups on protein **surfaces than their cationic counterparts.**^[12] Therefore, water-anion clusters represent an important chemical complex that is not yet fully understood despite their biological relevance. Several experimental and theoretical studies have been carried out to unmask the behaviour of hydrated halide clusters or other ions.^[13,14,15,16] Most of these clusters are formed by a single ion surrounded by a variable number of water molecules, although polyanionic clusters have also been reported.^[17] The recognition of two or more anions is critical to many biological processes, such in chloride transport.^[18] One important approach to study this type of clusters is by analysing those generated in confined spaces.^[19,20] Thus, several water clusters have been characterized by studying those formed in small protein pockets or other nanostructured materials. In this work we describe a new class of hydrated **polyanionic** clusters, **most likely three**, embedded between two cyclic peptide subunits of a novel supramolecular dimeric capsule.

Figure 1.

Supramolecular capsules are dynamic molecular assemblies composed of two or more self-complementary units held together by non-covalent interactions that, upon assembling, form an empty closed system whose internal cavities are useful tools in host–guest chemistry.^[21,22] They can be considered to create a new kind of molecular environment in which trapped molecule(s) exist. A wide variety of supramolecular capsules have been prepared with complementary or opposite functions, ranging from large to small, from charged to neutral or from open to closed systems. They exhibit unique properties, such as remarkable spectrochemical, electrochemical or magnetic effects. Therefore, they are useful tools for the molecular recognition of a wide variety of anionic, cationic or neutral guests (including gases) and are also able to catalyse a number of reactions leading to unusual products in a remarkably efficient way.^[23,24,25,26] Recently, we have shown that flat-shaped cyclic peptides^[27,28] equipped with a molecular cap (porphyrin moiety) provide a hydrogen-bonded dimeric receptor that was able to entrap long linear 4,4'-bipyridyl derivatives (Figure 1).^[29] Here, we have created a smaller variant topped with a *tris*-triazolylamine component that is able to recognize a new class of water-anion clusters. To do this, the amide protons involved in the hydrogen bonds that supramolecularly bind both receptor subunits participate in the recognition of the hydrated anion. These clusters sandwiched between the two cyclic peptide components resembles a “molecular burger”, in which the anionic complexes are intercalated, like meat patty and cheese, between the two subunits that play the role of a “molecular bun”. The isobutyl pendants of Leu side chain are cross-interdigitated providing a hydrophobic supramolecular complex that can facilitate ion transport across lipid membranes.

Results and Discussion. Recently, we have prepared the cyclic hexapeptide bearing three propargyl chains attached to the γ -residues (**CPI**) that are pointing opposite to the dimerization plane (**D1**).^[30,31,32] These groups not only prevent stacking of cyclic peptides across that face, thereby restricting the assembling process to discrete

dimers rather than nanotubes, but can also act as valuable reactive element for the incorporation of other functional moieties at the entrance of the dimer (Figure 1). For example, Sonogashira cross-coupling provided derivatives bearing ortho- or meta-oriented pyridine moieties.^[31] The resulting peptides were able not only to encapsulate Xe atoms in the tubular structure of stacked dimers, but also to efficiently transport ion pairs in model membranes.^[30] Additionally, the acetylene moiety was used in a copper-catalysed alkyne-azide cycloaddition reaction to incorporate a variety of oligoethylene glycol pendants to generate novel membrane spanners.^[32] Here, we describe the preparation of self-assembled cavitands (Figure 2) covered by a *tris*-triazolyl motif attached to a central core that not only plays the role of a molecular cap but also acts as a binding site for the recognition of small anions.^[33, 34,35,36,37,38]

Figure 2.

Taking in account the aforementioned objectives, the cyclic peptide **CP1** was synthesized using the protocol described previously (Figure 2).^[30] For the cap incorporation, **CP1** was subjected to a triple click reaction using the previously optimised method in which catalytic amounts of air-sensitive copper complex [Cu(MeCN)₄PF₆] (10 mol% per alkyne) were used in presence of DIEA and TBTA ligand.^[32,39] Under these conditions the expected cavitand **CP2** was obtained, although in low yield (15%). After screening for improving reaction conditions, a new protocol was found using copper bromide and DBU in toluene, providing **CP2** in 48% yield.^[40]

Figure 3.

This peptide retained the self-assembling properties of its precursor, **CP1**, adopting the proposed flat conformation and forming the C₃-symmetric dimer (**D2**), as denoted by NMR experiments (**Supplementary Fig. 1**).^[41,42] Clear evidence of dimer formation was derived from the down-field shift of the amide proton (8.04 ppm) whose

chemical shift did not change with CP dilution up to 200 μM in CD_2Cl_2 , confirming the large dimerization constant ($K_a > 10^4 \text{ M}^{-1}$) similar to the non-capped derivatives.^[29] Further evidence of the dimeric structure derived from the MS (2045.25 $[\text{M}_{\text{D2}}+\text{Na}]^+$, **Supplementary Fig. 2**) and FTIR spectroscopy, with amide A band a 3286 cm^{-1} and carbonyl vibrations at 1623 and 1532 cm^{-1} , characteristic of the antiparallel-type β -sheet structure (**Supplementary Fig. 3**).^[43,44,45] The final evidence was obtained from the X-ray crystallography analysis of a crystal obtained from a solution of **D2** in 10% CD_3CN in CD_2Cl_2 from which a dimeric supramolecular capsule held by six hydrogen bonds between both subunits was found (Figure 3a-b). Interestingly one acetonitrile moiety is entrapped in the inner cavity of this dimer, with its nitrile group pointing towards the protons of the triazolyl moieties of one of the subunits. On the other side, one of the triazole moieties is rotated, arranging two of the nitrogen atoms of the heterocycle towards the internal cavity, perhaps with the goal of partially filling it due to the lack of a suitable guess. We checked if *bis*-nitrile derivatives, i.e., malononitrile (MN), could interact simultaneously with both *tris*-triazole motives improving its encapsulation (**Supplementary Figs. 4, 5 and 6**). We found out that it was necessary to add up to 30 equivalents of malononitrile (**Supplementary Fig. 5**) to a dichloromethane (CD_2Cl_2) solution of **D2** to get a new species (MNC_{D2}), in a 4 to 1 ratio with respect to the empty capsule. This new species was assigned to the encapsulated complex due to the down-field shift of triazole proton ($\Delta\delta = 0.06 \text{ ppm}$). 2D-NMR experiments confirmed that $\text{H}_{\beta_{\text{Acp}}}$ (*cis*-oriented with carboxy and amino groups) suffered down-field shift (**Supplementary Fig. 6**), which is consistent with the incorporation of a malononitrile molecule in the cavity. The weak interaction is most likely due to the 109° angle between the two nitriles, which possibly hinders the simultaneous formation of strong hydrogen bonds with both *tris*(triazolyloethyl)amine moieties. Succinonitrile (SN) was also evaluated, considering its additional methylene moiety allows both cyano groups to be arranged at a 180° angle, although we were concerned that its length would exceed

the size of the cavity. Fortunately, the addition of small amounts of SN (~0.5 equiv) over a CD₂Cl₂ solution of **D2** (6.4 mM) already gave rise to a new set of signals corresponding to **SN⊂D2** (Supplementary Figs. 4 and 7), suggesting a higher affinity than MN. After the addition of 7 equiv, the encapsulated complex (**SN⊂D2**) is already the main component in the mixture, although in a 1.4:1 ratio with **D2** (Supplementary Fig. 7A). Whereas for the MN recognition there is almost no change in the chemical shift of amide protons, for the SN the signal is up-field shifted (7.84 ppm) suggesting a tight binding that reduced the conformational freedom (signals are sharper than those of the free capsule) and stress the hydrogen bond network. The new singlet at 3.30 ppm corresponds to the methylene groups from the entrapped SN molecule as indicated the 2D-NMR experiments (Figure S7B-C). DFT geometry optimizations confirm the stability of both complexes (Supplementary Fig. 8, see Supplementary Discussion III for further information). The main difference between both encapsulated *bis*-nitrile complexes was the length of hydrogen bonds, which become longer as the size of the encapsulated nitrile increases.

Figure 4.

Once the ability of **D2** to form hydrogen-bonded supramolecular inclusion complexes with appropriate guests and the capability of proton of *tris*-triazole cap to form hydrogen bonds with the entrapped molecule was confirmed, the recognition of small anions was evaluated. Initially, we assessed halide recognition using NMR titrations (specifications regarding ¹H NMR titrations are detailed in Supplementary Discussion I). Clearly, addition of different portions of tetrabutylammonium fluoride to a solution of **CP2** (7.3 mM) in 10% CD₃CN/CD₂Cl₂ gave rise to a new set of signals that were attributed to the recognition of the fluoride anion (Figure 4). In any case, it was necessary to add ca. 3 equivalents of TBAF to achieve the complete disappearance of the signals corresponding to **D2**. Similar results were obtained with the addition of

TBACl (Supplementary Fig. 9) under similar conditions (10% CD₃CN/CD₂Cl₂), but in this case it was necessary to add more than 15 equivalents of chloride to fully shift the equilibrium towards the new species, suggesting a weaker interaction (Table 1). No changes were observed for the addition of larger halides (bromide and iodide) or nitrate. Careful analysis of the ¹H NMR spectra of both experiments reveal that, as opposed to what it was found in anion recognition with *tris*(triazolyl) receptors,^[33-38] the heterocyclic protons are up-field shifted (7.47 ppm). This suggests that these protons must not be participating in halide recognition. On the other hand, the amide proton signals are down-field shifted (from 8.00 to 10.00 ppm for fluoride and to 8.62 ppm for chloride), implying their participation in stronger hydrogen bonds. Conformational changes derived from the interaction with the anions are clearly evidenced by the splitting of the signal of the methylene linker (singlet at 4.59 ppm for **D2**) between the triazole rings and the cyclic peptide backbone into two doublets at 4.90 and 4.04 ppm. In addition, conformational changes at the CP backbone suggest changes in the flat conformation characteristic of the sheet structure. For example, H α _{Leu}, initially at 4.81 ppm, undergoes a remarkable up-field shift to 4.15 ppm for fluoride (4.19 for chloride), while the H γ _{Acp}, initially at 4.73 ppm, undergoes an opposite, albeit milder, down-field shift to 4.90 ppm (4.83 ppm for chloride). Two-dimensional NMR experiments provided evidence of conformational changes, in which nOe cross-peaks that are not present in the empty capsule appear after addition of the halide (Supplementary Fig. 10), such as the one between H γ _{Acp} and one of the protons of H β _{Leu} upon recognition of fluoride. The disappearance, upon complexation, of the strong nOe cross-peaks between H_{triazole} and H α _{Leu} and of the methylene linker and the H β _{Leu} also seems relevant. Further evidence of the halide and CP interactions were found at ESI-MS, in which several ion peaks corresponding to complexes between both components are the main detectable species (Supplementary Fig. 11).

Table 1.

In a search for improving the binding, we discovered that water molecules play a key role in the supramolecular recognition process. The addition of 4Å molecular sieves, with the aim of drying the solution of TBACl and **D2**, gives rise to the recovery of the starting dimeric capsule. Filtration of the resulting solution and addition of 10 µL of water provoke the recovery of the chloride-capsule complex ($3\text{Cl}\cdot\text{nH}_2\text{O}\cdot 2\text{CP2}$) spectrum (Supplementary Fig. 12A). Inspired by this finding, new studies with the other larger and less hygroscopic anions were reviewed to evaluate the role of water molecules in their recognition. The addition of a few drops of water (7 µL) to the 10% $\text{CD}_3\text{CN}/\text{CD}_2\text{Cl}_2$ solution of **D2** (4.7 mM), resulted in only a small down-field shift of the triazole proton signal, presumably due to water encapsulation (Supplementary Fig. 13); but successive additions of increasing amounts of TBAB yielded similar changes to those already described for chloride additions (28 equivalents added and δ_{NH} 8.37 ppm, Table 1) (Supplementary Fig. 14). Remarkably, even recognition of iodide was also observed, although more equivalents of iodide (86 equiv, Table 1) were required, and the down-field shift of NH signal (8.07 ppm) was even smaller (Supplementary Fig. 15). This definitely underlines the importance of water molecules in the anion recognition process, most likely due to the ability of the capsule to recognize the hydration spheres of the anions.^[9] Next, other anions (see Table 1, Supplementary Figs. 16, 17, 18 and 19) were evaluated and a similar behaviour was found; those that were recognised, such as nitrate, acetate or azide ions, only did so in the presence of some water molecules. Larger ions, such as tribromide or hexafluorophosphate did not interact even in the presence of different amounts of water. These results confirm the remarkable ability of **D2** to recognise the hydration spheres of different anions. We decided to further study the recognition of acetate to stress out the importance of water molecules in the complexation process by drying this solution. After incubating a solution of **D2** containing 66 equivalents of TBAA with molecular sieves for 48 hours, the ^1H NMR signals corresponding to **D2** were recovered, while those assigned to the original

complex disappeared. The subsequent addition of few drops of water (10 μ L) immediately re-established the original complex (Supplementary Fig. 12B). Interestingly, while overnight incubation was sufficient to dehydrate the encapsulated chloride complex with molecular sieves, drying the acetate mixture required two days for the full recovery of **D2**, suggesting a tighter binding. Apart of the key role paid by water molecules, the chemical shift and number of equivalents of the different anions tested suggest that size, shape, and basicity are important parameters in the recognition process. To confirm the role played by the capsule and the *tris*(triazolyethyl)amine cap, **CP1** was evaluated for the recognition of anions. TBAF additions over a solution of **D1** in dichloromethane with a 10% of acetonitrile did not provide any change in its ^1H NMR spectrum, confirming the relevance of the cap in the anion/water cluster recognition (Supplementary Fig. 20). The role of the tetrabutylammonium counterion was also evaluated, for this purpose we decided to make use of the known affinity of crown ethers for alkali cations.^[46] The 15-crown-5 is known for being the one whose radius fits better with Na^+ .^[47] Therefore, we were able to solubilize NaOAc in deuterated acetonitrile using this crown ether and this solution was used in new titration experiments with **D2** in dichloromethane. Similar results were obtained, although larger amounts of the sodium acetate/crown ether acetonitrile solution were required to shift the equilibrium, most likely due to the lower solubility of the complex under these conditions (Supplementary Fig. 21A). In fact, the NMR spectrum of the mixture two weeks after titration showed a clear increase of the acetate complex with respect to the initial conditions, going from 1.4:1 ratio to 3:1 (Supplementary Fig. 21B). This can be attributed to the slow solubilization of the acetate salt triggered by the formation of the complex. ROESY spectra showed a similar cross-peaks pattern to those obtained using tetrabutylammonium as counterion (Supplementary Fig. 22).

The strong down-field shift of amide protons upon anion addition suggests their involvement in its recognition, consequently, further experiments were carried out to

understand this behaviour. It is well established that cyclic peptides made by five-membered ring γ -amino acids (Acp) form heterodimeric complexes with CPs containing six-membered ring γ -amino acids, i.e. **CP3**,^[42,48,49] which are more stable than the corresponding homodimers. Therefore, we decided to evaluate the importance of capsule integrity in the cluster recognition. Consequently, **CP3**, which was prepared following a similar strategy to the one used in the preparation of **CP2** but starting from 3-aminocyclohexanecarboxylic acid (Ach, **Supplementary Fig. 23**), also forms the corresponding dimer **D3** in dichloromethane solution. **CP3** was mixed with a solution of **D2**, and the resulting mixture showed in the NMR spectra the appearance of a new set of signals that were assigned to the supramolecular aggregate **D2-3** (Figure 5 and **Supplementary Fig. 24**). Further evidence of the heterodimeric structure derived from MS experiments (**Supplementary Fig. 25**) in which the 1840.18 ion peak corresponds to the heterodimeric complex. This confirmed, once again, the higher stability of the heterodimeric aggregates because of the better van der Waals fitting between both cycloalkyl rings.^[50] The formation of this heterodimer allowed us to evaluate if dimers containing only one *tris*(triazolyethyl)amine cap (cavitand **D2-3**) could still recognize anions. Interestingly, after the addition of small portions of fluoride or chloride (TBAF or TBACl) on the CD₂Cl₂ solution of heterodimer **D2-3**, simultaneous recovery of homodimer **D3** and the formation of the corresponding complexes of **CP2** with the anions (**3F⁻.nH₂O** and **3Cl⁻.nH₂O**) were observed. Three equivalents of fluoride were also required to shift the equilibrium to the homodimeric components. This confirms that the interaction with hydrated anions is even more favourable than that of the heterodimer and that both capped subunits are necessary in this process.

Figure 5.

Cyclic peptide **CP3** was then transformed into the *tris*(triazolyethyl)amine capped Ach derivative (**CP4**) using similar conditions in 50% yield (Figure 2).

Therefore, the recognition properties of the new Ach-based capsule topped with the *tris*(triazolyethyl)amine motif were also studied. These cavitand self-dimerized in dichloromethane solution to form **D4**, as denoted by NMR, MS and FTIR (see Supplementary Information section 7 and 8). Once again, single crystal suitable for X-ray diffraction of this compound also confirmed the capsule formation that also entrap one acetonitrile molecule in its cavity (Figure 3C-D). **D4** has all the triazole protons pointing towards the internal cavity making its structure more symmetric than the Acp-based one (figure 3a-b), with similar length in all the interpeptide hydrogen bonds. These are generally longer (2.24 Å) than those of **D2**, which range from 1.91 to 2.35 Å (2.10 Å in average), although the capsule **D4** is slightly more compact with a shorter distance between the two nitrogen atoms of the *tris*(triazolyethyl) caps (14.30 Å versus 14.85 Å). In contrast to the recognition properties of **D2**, additions of more than twenty equivalents of TBAF or TBACl do not result in any change in the NMR spectra of **D4** (Supplementary Fig. 26). This indicates that the Ach-based capsule is not capable of recognising anions, most likely due to the greater rigidity of the six-membered ring of this γ -amino acid that prevents the CP from adopting the appropriate conformation for the recognition of such species.

Both cavitands (**CP2** and **CP4**) also assemble into the heterodimer **D2-4** (Figure 5 and Supplementary Fig. 27A) when an equimolar mixture of both compounds in nonpolar solvents (CD_2Cl_2) is prepared. The appearance of new signals in the ^1H NMR spectra that do not correspond to any of the homodimers confirms the formation of **D2-4**. For example, the broad signal at 4.95 ppm belonging to the $\text{H}\alpha_{\text{Leu}}$ and the one at 4.22 ppm corresponding to one the methylene of the *tris*(triazolyethyl)amine cap are signals that belong to **CP2** of the heterodimer (Supplementary Fig. 27B). Moreover, the MS also confirms the formation of **D2-4** (Supplementary Fig. 28). Once again, addition of fluoride (TBAF) to this dichloromethane (CD_2Cl_2) mixture prompts the splitting of the heterodimer **D2-4** into the corresponding homodimer **D4** and the complex of hydrated

fluoride with cavitand **CP2** ($3\text{F}^- \cdot n\text{H}_2\text{O} \subset 2\text{CP2}$). In all cases, the addition of more than three equivalents of fluoride was necessary to achieve complete dissociation of the heterodimers. DFT geometry optimization of both heterodimeric species, **D2-3** and **D2-4**, are shown in Supplementary Fig. 29, which once again confirmed the stability of the mentioned dimers and provide further information about the hydrogen bonds length and angles (see also Supplementary Discussion III for further information).

Figure 6.

Crystals suitable for X-ray diffraction were obtained from solutions containing the supramolecular capsules **D2** and the tetrabutylammonium halides (chloride and fluoride, Figure 6). To our delight, in both cases the cyclic peptide capsules entrap a new kind of hydrated halide clusters made by three ions, as confirmed by the number of tetrabutylammonium ions that co-crystallized with the peptide capsule. Four and eight water molecules, for chloride and fluoride, respectively, are forming these clusters, confirming the higher tendency of the latter to have larger hydration shells. In both cases the three halide ions are distributed into six equivalent chemical positions that are shared with another three water molecules. Although for the chloride crystal all the ion positions are forming a hexagonal structure with all the positions placed at the same layer, for fluoride cluster, the six positions are placed at two different levels forming two triangular structures that are 60° rotated with respect to each other. To entrap these clusters, cyclic peptide dimers dissociate to allow amide protons to hydrogen-bond with halide ions and water molecules.^[51] It is notorious that in the solid state all the carbonyls are oriented towards the opposite side in which the interaction with the anionic cluster occurs, which could explain the observed up-field shift of $\text{H}\alpha_{\text{Leu}}$ in the NMR spectrum. This conformational change is due to the geometrical variations of the α -amino acids that go from the characteristic β -sheet conformation of flat disc-shaped CPs to a turn-

like structure. Interpeptide distance is slightly larger for the encapsulated fluoride cluster than that for chloride despite the larger size of the **latter**.

With respect to the fluoride-capsule complex (Figure 6a-b), the unit cell has two non-equivalent complexes (**3F⁻.8H₂O.2CP2**). In each complex, in addition to the three water molecules exchangeable by fluoride ions and hydrogen bonded to the **amide** proton, there are five other crystallographic positions preferably occupied by **water molecules**, although fluoride could also partially occupy any of these positions, since it is not possible to unambiguously differentiate both atoms due to their similar electron densities. In any case, to fulfil the hydrogen acceptor capability of the fluoride ion, we assume that these ions must be located in the positions in which they are bonded to the amide protons of the same cavitand and surrounded by three water molecules forming the first hydration shell of each fluoride (**Supplementary Fig. 30**). The fluoride occupancy in the two **3F⁻.8H₂O.2CP2** complex of each unit cell is not exactly the same (**Supplementary Fig. 31**), even though both complexes present analogue disposition. With respect to the rest of water molecules, there are two that are axially placed, forming hydrogen bonds with the fluoride-water exchangeable positions, while the other three are in the cluster equatorial perimeter forming hydrogen bonds with the exchangeable fluoride-water positions (**Supplementary Fig. 32**). These water molecules are placed at the position in which capsule is not fully closed forming a window. The Leu side chains are facing each other to create a hydrophobic oval-shaped aggregate that entrap the halide cluster in a non-polar environment.

Concerning the chloride complex (Figure 6c-d) the six equivalent positions are in the same plane and each chloride ion is hydrogen-bonded to one amide proton with N...Cl distance of 3.21 Å. In this case, the coating with the **leucine** side chains is less compact, leaving a wider window as compared to the fluoride complex. Furthermore, electron density can only be attributed to a maximum of four water molecules, one of

them located at 50% occupancy at the top and bottom of the hexagonal bipyramid and more deeply buried in the cavity of the supramolecular capsule, **remaining** partially hydrogen-bonded to the three chloride ions. In this crystal structure, the complex of the capsule with the chloride cluster is co-crystallized with the dimer **D3**, forming columns in which **D3** is alternated with the encapsulated trichloride cluster (**3Cl \cdot 4H₂O \subset 2CP2**). Within **D3** there is a dioxane molecule occupying three equivalent positions around the ternary symmetry axis of the dimer (**Supplementary Fig. 33**).

Unfortunately, it was not possible to obtain crystal structures of the complexes formed with the other anions (bromide, acetate, azide, iodide and so on) that would allow unequivocal confirmation of the formation of similar structures with these ions. In any case, the previous characterizations suggest the formation of clusters that are embedded in the equatorial cleft generated by the two CP subunits. To confirm this further, we carried out a detailed analysis comparing the NMR data of the different complexes and the X-ray diffraction data (**Supplementary Discussion II**), from which we concluded that the recognition process most likely involves the trapping by two **CP2** units of clusters composed of three anions, **although we do not have a conclusive evidence of such stoichiometry**, surrounded by several water molecules depending on the type of anion and its solvation. **To this complex we have used the coding mA⁻.nH₂O \subset 2CP2** for the complex with the anion cluster in which “*m*” represents the number of anions entrapped between both CPs (most likely three), “*A*” indicates the type of anion and “*n*” represents the number of water molecules that form each anion cluster.

After confirmation of the anion recognition ability of the supramolecular capsule **D2**, transmembrane transport experiments were carried out.^[1] For that purpose, lucigenin-trapped liposomes (**LG \subset LUV**) were prepared with which the intravesicular delivery of chloride ions was clearly established (Figure 7A).^[52] The transport is slow

compared with previously published chloride transporters and high concentration of capsule is required ($EC_{50} \sim 100 \mu\text{M}$). Additionally, we decided to examine the chloride transport ability of **D4** and **D1** using the lucigenin assay. As we expected from the previous NMR findings, neither **D1** nor **D4** were capable of transporting chloride (Supplementary Fig. 34), confirming that the recognition of the ions was necessary to be able to mediate its transport. Apart from that, further variations of the **D2** lucigenin assay, revealed that the transport efficiency did not change with the counterion used (Na^+ , Li^+ , K^+ or Cs^+), suggesting that cation is not involved in the transport process (Figure 7A3). To confirm the potential antiport transport of nitrate, experiments in which nitrate was substituted for the more hydrophilic sulphate, whose transmembrane transport is extremely difficult, were carried out (Supplementary Fig. 35).^[53,54] These experiments did not show any significant reduction in chloride transport rates, suggesting that chloride/nitrate exchange must not be involved in the transport mechanism. Therefore, the symport (H^+/Cl^-) or antiport (OH^-/Cl^-) must be associated to this migration. To confirm association of chloride transport with change in the pH, HPTS loaded vesicles (**HPTS \subset LUV**) were used (Figure 7B).^[55] Thus, vesicles basification promoted by **D2**, denoted by a fluorescence increase, would be associated to the co-transport of chloride. Unambiguous and fast enhancement of dye emission was found upon creating a pH gradient of almost one unit after the addition of a sodium hydroxide solution to the extravesicular medium, yielding an enhanced activity with $EC_{50} = 3 \mu\text{M}$, suggesting that anion transport was the rate limiting step and not the H^+ or OH^- co-transport. To clarify this finding, HPTS studies in the presence of a proton transporter (FCCP) were carried out (Figure 7D).^[56] As expected no changes in transport rates were found, confirming proton transport was not the limiting step. DLS measurements were then performed, which confirmed both the homogeneity and integrity of the vesicles throughout these experiments and, consequently, the observed

fluorescence changes are not due to bleaching or membrane disruption (Supplementary Fig. 36).

Finally, to evaluate the relative rates in anion transport, competitive experiments were carried out using **HPTS****C****LUV** (Figure 7C).^[52,57] For this purpose, vesicles whose internal buffer contained sodium chloride (100 mM) at pH 7 were placed in a variety of isosmotic buffer solutions with different sodium salts with other counterions. In this type of experiments, a pH gradient is generated if the transporter (**D2**) facilitates dominant ion influx or efflux depending on anion selectivity. These permeability differences give rise to a membrane potential that drives net proton transport. Vesicle acidification occurs when ion influx is faster than chloride efflux, while ions that are transported more slowly than chloride cause an increase in intravesicular pH. We found that acetate, fluoride, and azide provided vesicle basification, while bromide and iodide were transported faster than chloride, being acetate the slower influxed anion and iodide the faster one. Therefore, **D2** showed Hofmeister pattern ($I^- > Br^- > NO_3^- \sim Cl^- > N_3^- > F^- > OAc^-$) with the exception of nitrate that is almost as fast as chloride.^[58] The strongly hydrated, salting-out ions, are those with slower transport rates while previous ion recognition experiments showed the need of water molecules in the recognition of anion cluster and the strong binding for fluoride or acetate compared with iodide or bromide. Therefore, it seems that ions exchange (K_{on}/K_{off}) at the interfaces must play an important role in the transport process and not so much the selectivity of the binding itself. All the details regarding the transport assays are carefully addressed in Supplementary Discussion IV.

Figure 7.

In conclusion, we have presented the design, synthesis, and anion recognition properties of a cyclic α,γ -hexapeptide equipped with a *tris*(triazolyethyl)amine cap. We

have shown that, in presence of some anions, the self-assembled CP is able to reaccommodate its backbone conformation from a self-dimerizing flat circular-shaped structure to a triangular one that creates a binding pocket suitable for hosting hydrated anions. Moreover, we have rigorously identified and **studied** the different aspects involved in the recognition. First, the essential role played by the water molecules that are involved not only in the formation of anion-water clusters, forming the hydration shell of the different anions, and helping them to take the appropriate size and shape, but also in the interaction and entrapment inside the cavity formed by the two peptide hemicapsules. In addition, two key structural components of the cyclic peptide were identified: the five-member ring γ -amino acid and the *tris*(triazolyethyl)amine cap, without which the anion accommodation is not possible. Furthermore, we have shown that the cationic counterion does not play any relevant role in the supramolecular recognition, which supports our theory of the three main actors: **capped** peptide, water and anion.

Finally, the anion transporting properties **were** also explored. Our studies have shown that **D2** can transport different anions across model lipid membranes, with a tendency to favour **the** transport of anions with weaker hydration spheres, which are generally **weaker** recognized. Transport is associated with the modification the pH of the intravesicular medium, most likely through a Cl^-/H^+ symport mechanism, although the Cl^-/OH^- antiport exchange could not be ruled out, the rate-limiting step being the migration of the anions. The peptide composition, design simplicity and anion transporting properties linked to pH regulation activity open-up several avenues for **the** therapeutic use of these novel self-assembled systems.^[59]

Methods

General ^1H -NMR titrations protocol

Stock solutions of internal standard, either dioxane (2.5 mM) or TMSS (0.15 mM), in a mixture of CD₃CN in CD₂Cl₂ (10 % v/v) were prepared. These solutions were used to prepare all the samples needed in the titration experiments, in order to keep the concentration of the internal standard constant throughout the experiments. The signals from both standards, dioxane (s, 3.6 ppm in CD₂Cl₂) or TMSS (s, 0.18 ppm in CD₂Cl₂), were used to calculate the concentration of all the components along the titration experiments.

Generally, 450 µL of the solution containing the corresponding peptide capsule (**D2**, **D3** or **D4**, ca. 5mM) was placed in the NMR tube. After recording the ¹H NMR spectra of the starting sample, successive additions of the corresponding titrant were made via micropipette and the corresponding spectrum was taken.

General procedure for vesicles preparation

Vesicles were prepared, under argon, in a round bottom flask by slow evaporation of a solution of EYPC in CHCl₃ (25 mg/mL, 1 mL) to form a thin and homogeneous film on the flask surface. The film was dried overnight under high vacuum and then carefully hydrated with the intravesicular aqueous media. The resulting mixture was **tumbled** for 1 hour in the rotavapor at 180 r.p.m. but at atmospheric pressure. Every 15 minutes, the rotavapor rotation angle was changed (60°, 50°, 45° and 35°) in order to get and homogeneous dispersion. After that, the milky sample was subjected to 11 freeze-thaw cycles (N₂ (l) → 40 °C water), and the resulting suspension was extruded (25 times) across polycarbonate membrane (200 nm pore size). Finally, the suspension was passed through a size exclusion column (Sephadex G-25) previously equilibrated with the isosmotic extravascular medium.^[56] The resulting vesicle suspension was taken up in a total volume of 5 mL, giving an approximate lipid concentration 6.6 mM. Size and **particle** number consistency between different vesicles batches were checked using DLS.

General procedure for transport measurement

In a plastic cuvette containing a 4 mm diameter stirring bar, the previously prepared vesicle suspensions (50 μL) were dispersed in the extravesicular media (1950 μL). The cuvette was placed with moderate stirring in the fluorometer equipped with a module that allows the successive additions of titrant during the measurement **in the dark**. Data (fluorescence emission band at 535 nm for lucigenin essays or 510 nm for HPTS experiments) were collected for thirty minutes every second. One minute after the experiment started, aqueous solutions of NaCl (25 μL , 2 M, lucigenin assay) or NaOH (25 μL , 0.5 M, HPTS assay) were added. After an additional minute (minute 2 of the measurement), a solution of the CP at different concentrations (25 μL , in iPrOH for the lucigenin assay or in DMSO for the HPTS) were added. Finally, after 27 minutes an aqueous solution of Triton (50 μL , 10% v/v) was added to provoke **liposome lysis**. After three more minutes of stabilization, the resulting signal was used for the data normalization.

DLS measurements

For each batch of vesicle prepared, some control measurements were performed before the transport experiments to check the homogeneity between batches. The size, polydispersity index and particle concentration were measured using the same volumes **as** in the fluorescence assay. Four samples were collected and checked for each essay. Regarding the lucigenin experiments (**Supplementary Fig. 36**), the first sample, containing only the vesicles dispersed in the extravesicular media (blue line), the second sample, which corresponds to the mixture after the addition of the aqueous solution of NaCl (green line), the third measurement was carried out after the incorporation of CP (pink line), and finally, the fourth sample was done after the addition of Triton X-100 (red line). The four samples were stirred for 25 minutes, **and after a 5 min lag resting**

perioded the DLS were recorded. For the HPTS assay (See Supplementary Fig. 36) in HEPES buffer (10 mM, NaCl 100 mM, pH 7), the analyzed mixtures correspond to the initial conditions (blue line), and after successive additions of NaOH (green line), cyclic peptide (pink line) and Triton (red line).

Notice that samples measurement indicates monodispersity, in view of the sharp slope of the correlation curves, which was maintained in all samples except after the addition of the surfactant, Triton X-100, when the vesicles undergo lysis. After lysis, the faster decay of the correlation coefficient is a signal of the smaller particle size. Also, the slope, less sharp, indicates higher polydispersity. We measured the particle concentration, which also indicated homogeneity between batches and confirmed the stability of the vesicles until **their** lysis.

DFT geometry optimizations

Computational studies were carried out using the sources from Centro de Supercomputación de Galicia (CESGA). All calculations were performed with the Gaussian 16 rev. C01 package. The geometries used in the calculations were based on the crystal structures derived from this study. Calculations were performed in vacuum. We carried out DFT geometry optimizations using B3LYP with the moderate-size basis set 6-31G (d, p). We also included GD3BJ as empirical dispersion.

Supplementary Information accompanies the paper on <https://www.nature.com/nchem/>. Detailed descriptions of the synthesis and characterization of key compounds, including NMR spectra (^1H and ^{13}C , NOESY and/or ROESY) and FTIR spectra of peptides **CP2**, **CP3** and **CP4**. CCDC-2311116 to CCDC-2311119 contains the supplementary crystallographic data for this paper. The data can be obtained free of charge from The Cambridge Crystallographic Data Centre via <https://www.ccdc.cam.ac.uk/structures>.

Acknowledgements This work was supported by the Spanish Agencia Estatal de Investigación (AEI) (PID2019-111126RB-100 and PID2022-142440NB-I00), the Xunta de Galicia (ED431C 2021/21 and Centro singular de Investigación de Galicia accreditation 2019-2022, ED431G 2019/03), and the European Union (European Regional Development Fund - ERDF). We also thank the ORFEO-CINCA network and Mineco (RED2022-134287-T). V.L.-C. thanks the Xunta de Galicia for her research contract (ED481A-2019/117). All calculations were carried out at the Centro de Supercomputación de Galicia (CESGA).

Correspondence and requests for materials should be addressed to Juan R. Granja (juanr.granja@usc.es).

Figure 1. Previously reported supramolecular containers based on α,γ -cyclic peptides. Top: cyclic octapeptide (blue) topped with a porphyrin moiety (green) used in the recognition of 4,4'-bipyridines. Center: smaller alternatives derived from dimer-forming *N*-propargylated cyclic hexapeptides (CPI, grey) through Sonogashira cross-couplings with Iodopyridines or copper-catalyzed azide-alkyne cycloaddition (CuAAC) that have been used as ion transporters.^[27] Bottom: cartoon model of supramolecular capsule derived from CPI and a *tris*-azide derivative described in this work.

Figure 2. Synthetic strategy used for the preparation of capsules **D2** and **D4** and initially proposed encapsulation model for the recognition of **anions** (**2XC2** and **2XC4**).

Figure 3. Side (a and c) and top (b and d) views of the crystal structures of dimeric supramolecular capsules **D2** (top) and **D4** (bottom), respectively. The molecules of acetonitrile entrapped in the cavity are represented in CPK models. The nitrogen of nitrile groups is pointing towards one of the caps close to the triazole protons with shorter distance in the Ach-based capsule (2.66 Å, bottom) than in the Acp derivative (2.79-2.68 Å, top). For clarity only triazole and amide protons are shown. The yellow dashed lines highlight the hydrogen bonds.

Figure 4. NMR spectra of pure **CP2** (bottom) and after the addition of different equivalents of fluoride (TBAF). In blue colour are highlighted the signals corresponding to the new species formed after the addition of the fluoride, light blue denotes the signals corresponding to the tetrabutylammonium counterion.

Figure 5. Experiments of heterodimer (**D2-3** and **D2-4**) formation followed by anion recognition, top view. Bottom a) NMR spectra corresponding to these studies in which the characteristic signals of each component are highlighted with specific colours; orange, green and lavender for homodimers **D2**, **D3** and **D4**, respectively, dark blue for **CP2** interacting with fluoride, and plum and teal green for heterodimers **D2-3** and **D2-4**, respectively.

Figure 6. Side (a) and top (b) view of the crystal structure of water-fluoride cluster entrapped in supramolecular capsule **D2** ($3\text{F}\cdot 8\text{H}_2\text{O}\subset 2\text{CP2}$) as *twin* complexes. In these complexes three fluoride ions (light green) and eight water molecules are hydrogen-bonded to the amide protons of two cyclic peptides at different planes (the hydrogen bond network (yellow dashed lines) in the cluster is only shown for one of the complexes). Side (c) and top (d) view of the crystal structure of encapsulated chloride-water cluster between two **CP2** ($3\text{Cl}\cdot 4\text{H}_2\text{O}\subset 2\text{CP2}$). The three chloride ions and water molecules are occupying six equivalent chemical (and crystallographic) positions forming a hexagonal structure, where the two subunits are aligned forming a trigonal bipyramid shaped (d) capsule.

Figure 7. Chloride transport experiments using capsule **D2** in liposomes containing lucigenin (A) or HPTS (B-D).

Table 1. Key features of the anion encapsulation by *tris*-triazoyl modified CPs (**D2** and **D4**); On the right: illustrative anion binding experiments derived by titrations carried out by NMR experiments. The dashed lines are used to indicate the anions in which extra water was added to facilitate complex formation. ^[a] equivalent number are given with respect to CP2 concentration, ^[b] Molar fraction was calculated at the mentioned maximum number of equivalents of the corresponding anions, ^[c] 7 μ L of water were added before the additions of anion solution.

[1] Davis, J. T.; Gale, P. A. & Quesada, R. Advances in anion transport and supramolecular medicinal chemistry. *Chem. Soc. Rev.* 49, 6056–6086 (2020).

[2] Rowe, S. M.; Miller, S. & Sorscher, E. J. Cystic fibrosis. *N. Engl. J. Med.* 352, 1992–2001 (2005).

[3] McNaughton, D. A.; Fares, M.; Picci, G.; Gale & P. A. Caltagirone, C. Advances in fluorescent and colorimetric sensors for anionic species. *Coord. Chem. Rev.* 427, 213573 (2021).

[4] Chen, L.; Berry, S. N.; Wu, X.; Howe, E. N. W. & Gale, P. A. Advances in anion receptor chemistry. *Chem* 6, 61–141 (2020).

[5] Molina, P.; Zapata, F. & Caballero, A. Anion recognition strategies based on combined noncovalent interactions. *Chem. Rev.* 117, 9907–9972 (2017).

-
- [6] Liu, Y.; Sengupta, A.; Raghavachari, K. & Flood, A. H. Anion binding in solution: Beyond the electrostatic regime. *Chem* 3, 411–427 (2017).
- [7] Stockbridge, R.; Kolmakova-Partensky, L.; Shane, T.; Koide, A.; Koide, S.; Miller, C. & Newstead, S. Crystal structures of a double-barrelled fluoride ion channel. *Nature* 525, 548–551 (2015).
- [8] Wu, X.; Wang, P.; Turner, P.; Lewis, W.; Catal, O.; Thomas, D. S. & Gale, P. A. Tetraurea macrocycles: Aggregation-driven binding of chloride in aqueous solutions. *Chem* 5, 1210–1222 (2019).
- [9] Gomez, D. T.; Pratt, L. R.; Asthagiri, D. N. & Rempe, S. B. Hydrated anions: From clusters to bulk solution with quasi-chemical theory. *Acc. Chem. Res.* 55, 2201–2212 (2022).
- [10] Balamurugan, K. & Pisabarro, M. T. Stabilizing Role of Water Solvation on Anion- π Interactions in Proteins. *ACS Omega* 6, 25350–25360 (2021).
- [11] Varma, S.; Sabo, D. & Rempe, S. B. K⁺/Na⁺ Selectivity in K Channels and Valinomycin: Over coordination Versus Cavity-size constraints. *J. Mol. Biol.* 376, 13–22 (2008).
- [12] Yao, W.; Wang, K.; Wu, A.; Reed, W. F. & Gibb, B. C. Anion binding to ubiquitin and its relevance to the Hofmeister effects. *Chem. Sci.* 12, 320–330 (2021).
- [13] Castleman, A. W. & Keesee, R. G. Ionic clusters. *Chem. Rev.* 86, 589–618 (1986).
- [14] Jungwirth, P. & Cremer, P. Beyond Hofmeister. *Nat. Chem.* 6, 261–263 (2014).
- [15] Marcus, Y. Effect of ions on the structure of water: Structure making and breaking. *Chem. Rev.* 109, 1346–1370 (2009).
- [16] Cremer, P. S.; Flood, A. H.; Gibb, B. C. & Mobley, D. L. Collaborative routes to clarifying the murky waters of aqueous supramolecular chemistry. *Nat. Chem.* 10, 8–16 (2018).
- [17] He, Q.; Tu, P. & Sessler, J. L. Supramolecular chemistry of anionic dimers, trimers, tetramers, and clusters. *Chem* 4, 46–93 (2018).

-
- [18] Harris, A.; Saita, M.; Resler, T.; Hughes-Visentin, A.; Maia, R.; Pranga-Sellnau, F.; Bondar, A.-N.; Heberle, J. & Brown, L. S. Molecular details of the unique mechanism of chloride transport by a cyanobacterial rhodopsin. *Phys.Chem.Chem.Phys.* 20, 3184–3199 (2018).
- [19] Sokkalingam, P.; Shraberg, J.; Rick, S. W. & Gibb, B. C. Binding hydrated anions with hydrophobic pockets. *J. Am. Chem. Soc.* 138, 48–51 (2016).
- [20] Chakraborty, S.; Dutta, R.; Wong, B. M. & Ghosh, P. Anion directed conformational diversities of an arene based hexa-amide receptor and recognition of the $[F_4(H_2O)_6]^{4-}$ cluster. *RSC Adv.* 4, 62689–62693 (2014).
- [21] Dariush, A. Self-assembled molecular capsule. In *Synergy in Supramolecular Chemistry*; Tatsuya, N., Ed.; CRC Press: Boca Raton, FL, 2015; pp 133–148.
- [22] Solomonov, A. & Shimanovich, U. Self-assembly in protein-based bionanomaterials. *Israel J. Chem.* 60, 1152–1170 (2020).
- [23] Saha, R.; Mondal, B. & Mukherjee, P. S. Molecular cavity for catalysis and formation of metal nanoparticles for use in catalysis. *Chem. Rev.* 122, 12244–12307 (2022).
- [24] McTernan, C. T.; Davies, J. A. & Nitschke, J. R. Beyond platonic: How to build metal–organic polyhedra capable of binding low-symmetry, information-rich molecular cargoes. *Chem. Rev.* 122, 10393–10437 (2022).
- [25] Bowman-James, K. Supramolecular cages trap pesky anions. *Science* 365, 124–125 (2019).
- [26] Rizzuto, F. J.; von Krbek, L. K. S. & Nitschke, J. R. Strategies for binding multiple guests in metal–organic cages. *Nat. Rev. Chem.* 3, 204–222 (2019).
- [27] Rodríguez-Vázquez, N.; Amorín, M. & Granja, J. R. Recent advances in controlling the internal and external properties of self-assembling cyclic peptide nanotubes and dimers. *Org. Biomol. Chem.* 15, 4490–4505 (2017).

-
- [28] Song, Q.; Cheng, Z.; Kariuki, M.; Hall, S.C.L.; Hill, S.K.; Rho, J.Y. & Perrier, S. Molecular self-assembly and supramolecular chemistry of cyclic peptides. *Chem. Rev.* *121*, 13936–13995 (2021).
- [29] Ozores, H. L.; Amorín, M. & Granja, J. R. Self-assembling molecular capsules based on α,γ -cyclic peptides. *J. Am. Chem. Soc.* *139*, 776–784 (2017).
- [30] Pizzi, A.; Ozores, L. H.; Calvelo, M.; García-Fandiño, R.; Amorín, M.; Demitri, N.; Terraneo, G.; Bracco, S.; Comotti, A.; Sozzani, P.; Bezuidenhout, C. X.; Metrangolo, P. & Granja, J. R. Tight Xenon confinement in a crystalline sandwich-like hydrogen-bonded dimeric capsule of a cyclic peptide. *Angew. Chem. Int. Ed.* *58*, 14472–14476 (2019).
- [31] Fuertes, A.; Amorín, M. & Granja, J. R. Versatile symport transporters based on cyclic peptide dimers. *Chem. Commun.* *56*, 46–49 (2020).
- [32] Fuertes, A.; Amorín, M. & Granja, J. R. Membrane spanners based on a dimeric α,β -cyclic peptide core. *Supramol. Chem.* *32*, 239–246 (2020).
- [33] Lee, C.-H.; Lee, S.; Yoon, H. & Jang, W.-D. Strong binding affinity of a Zinc–porphyrin-based receptor for halides through the cooperative effects of quadruple C–H hydrogen bonds and axial ligation. *Chem. Eur. J.* *17*, 13898–13903 (2011).
- [34] Parks, F.C.; Liu, Y.; Debnath, S.; Stutsman, S. R.; Raghavachari, K. & Flood, A. H. Allosteric control of photofoldamers for selecting between anion regulation and double-to-single helix switching. *J. Am. Chem. Soc.* *140*, 17711–17723 (2018).
- [35] Kim, H.; Hong, K.-I.; Lee, J. H.; Kang, P.; Choi, M.-G. & Jang, W.-D. Triazole-bearing calixpyrroles: strong halide binding affinities through multiple N–H and C–H hydrogen bonds. *Chem. Commun.* *54*, 10863–10865 (2018).
- [36] Liu, Y.; Zhao, W.; Chen, C.-H. & Flood, A. H. Chloride capture using a C–H hydrogen-bonding cage. *Science* *365*, 159–161 (2019).

-
- [37] Liu, Y.; Parks, F. C.; Sheetz, E. G.; Chen, C.-H. & Flood, A. H. Polarity-tolerant chloride binding in foldamer capsules by programmed solvent-exclusion. *J. Am. Chem. Soc.* *143*, 3191–3204 (2021).
- [38] Mondal, D.; Ahmad, M.; Panwaria, P.; Upadhyay, A. & Talukdar, P. Anion recognition through multivalent C–H hydrogen bonds: Anion-induced foldamer formation and transport across phospholipid membranes. *J. Org. Chem.* *87*, 10–17 (2022).
- [39] Fuertes, A.; Ozores, H. L.; Amorín, M. & Granja, J. R. Self-assembling Venturi-like peptide nanotubes. *Nanoscale* *9*, 748–753 (2017).
- [40] Bock, V. D.; Perciaccante, R.; Jansen, T. P.; Hiemstra, H. & van Maarseveen, J. H. Click Chemistry as a Route to Cyclic Tetrapeptide Analogues: Synthesis of cyclo-[Pro-Val- ψ (triazole)-Pro-Tyr]. *Org. Lett.* *8*, 919–922 (2006).
- [41] Amorín, M.; Castedo, L. & Granja, J. R. New cyclic peptide assemblies with hydrophobic cavities: The structural and thermodynamic basis of a new class of peptide nanotubes. *J. Am. Chem. Soc.* *125*, 2844–2845 (2003).
- [42] Brea, R. J.; Amorín, M.; Castedo, L. & Granja, J. R. Methyl-blocked dimeric α,γ -peptide nanotube segments: Formation of a peptide heterodimer through backbone-backbone interactions. *Angew. Chem. Int. Ed.* *44*, 5710–5713 (2005).
- [43] Haris, P. I. & Chapman, D. The conformational analysis of peptides using Fourier Transform IR spectroscopy. *Biopolymers (Pept. Sci.)* *37*, 251–263 (1995).
- [44] Krimm, S. & Bandekar, J. in *Advances in Protein Chemistry*, ed. C. B. Anfinsen, J. T. Edsall and F. M. Richards, Academic Press, Orlando, FL, 1986, pp. 181–364.
- [45] Barth, A. & Zscherp, C. What vibrations tell us about proteins. *Q. Rev. Biophys.* *35*, 369–430 (2002).
- [46] In-Hou, C.; Hong, Z. & Dearden V, David. Macrocyclic chemistry in the gas phase: intrinsic cation affinities and complexation rates for alkali metal cation complexes of crown ethers and glymes. *J. Am. Chem. Soc.* *115*, 5736–5744 (1993).

-
- [47] Buchanam, W. G.; Gerzain, M. & Bensimon, C. 1,4,7,10,13-Pentaoxacyclopentadecane (15-crown-5) sodium iodide complex, $C_{10}H_{20}O_5 \cdot NaI$. *Acta. Cryst.* **50**, 1016–1019 (1994).
- [48] Brea, R. J.; Pérez-Alvite, M. J.; Panciera, M. Mosquera, M.; Castedo, L. & Granja, J. R. Highly efficient and directional homo- and heterodimeric energy transfer materials based on fluorescently derivatized α,γ -cyclic octapeptides. *Chem. Asian J.* **6**, 110–121 (2011).
- [49] Aragay, G.; Ventura, B.; Guerra, A.; Pintre, I.; Chiorboli, C.; García-Fandiño, R.; Flamigni, L.; Granja, J. R. & Ballester, P. Self-sorting of cyclic peptide homodimers into a heterodimeric assembly featuring an efficient photoinduced intramolecular electron-transfer process. *Chem. Eur. J.* **20**, 3427–3438 (2014).
- [50] García-Fandiño, R.; Castedo, L.; Granja, J. R. & Vázquez, S. A. Interaction and dimerization energies in methyl-blocked α,γ -peptide nanotube segments. *J. Phys. Chem. B* **114**, 4973–4983 (2010).
- [51] Dutta, R. & Ghosh, P. Recent developments in anion induced capsular self-assemblies. *Chem. Commun.* **50**, 10538–10554 (2014).
- [52] Wu, X. & Gale, P. A. Measuring anion transport selectivity: a cautionary tale. *Chem. Commun.* **57**, 3979–3982 (2021).
- [53] Marcus, Y. Thermodynamics of solvation of ions. Part 5.—Gibbs free energy of hydration at 298.15 K. *J. Chem. Soc., Faraday Trans.* **87**, 2995–2999 (1991).
- [54] Tong, C. C.; Quesada, R.; Sessler, J. L. & Gale, P. A. *meso*-Octamethylcalix[4]pyrrole: an old yet new transmembrane ion-pair transporter. *Chem. Commun.* 6321–6323 (2008).
- [55] Kano, K. & Fendler, J. H. Pyranine as a sensitive pH probe for liposome interiors and surfaces. pH gradients across phospholipid vesicles. *Biochim. Biophys. Acta* **509**, 289–299 (1978).
- [56] Wu, X.; Small, J. R.; Cataldo, A.; Withcombe, A. M.; Turner P. & Gale, P. A. Voltage-switchable HCl transport enabled by lipid headgroup–transporter interactions. *Angew. Chem. Int. Ed.* **58**, 15142–15147 (2019).

[⁵⁷] Madhavan, N.; Robert, E. C. & Gin, M. S. A highly active anion-selective aminocyclodextrin ion channel. *Angew. Chem. Int. Ed.* **44**, 7584–7587 (2005).

[⁵⁸] Kang, B.; Tang, H.; Zhao, Z. & Song, S. Hofmeister series: Insights of ion specificity from amphiphilic assembly and interface property. *ACS Omega* **5**, 6229–6239 (2020).

[⁵⁹] Jowett, L. A.; Howe, E. N. W.; Soto-Cerrato, V.; Van Rossom, W.; Pérez-Tomás, R. & Gale, P. A. Indole-based perenosins as highly potent HCl transporters and potential anti-cancer agents. *Sci. Rep.* **7**, 9397 (2017).

[⁵⁶] Gilchrist, A. M. *et al.* Supramolecular methods: the 8-hydroxypyrene-1,3,6-trisulfonic acid (HPTS) transport assay. *Supramol. Chem.* **33**, 325–344 (2021).

REVIEWERS' COMMENTS

Reviewer #1 (Remarks to the Author):

Reviewer #1 was asked to look over the response given to Reviewer #2]

I feel the authors have successfully responded to the reviewers' comments and the manuscript is ready for publication.